Manuscript prepared for Hydrol. Earth Syst. Sci.
with version 2015/11/06 7.99 Copernicus papers of the LaTeX class copernicus.cls.
Date: 6 April 2016

# Global root zone storage capacity from satellite-based evaporation

Lan Wang-Erlandsson[1,2], Wim G. M. Bastiaanssen[2,3], Hongkai Gao[2,4],
Jonas Jägermeyr[5], Gabriel B. Senay[6], Albert I. J. M. van Dijk[7,8], Juan
P. Guerschman[8], Patrick W. Keys[1,9], Line J. Gordon[1], and Hubert H. G. Savenije[2]

[1]Stockholm Resilience Centre, Stockholm University, Stockholm, Sweden
[2]Department of Water Management, Faculty of Civil Engineering and Geosciences, Delft
University of Technology, Delft, the Netherlands
[3]UNESCO-IHE Institute for Water Education, Delft, The Netherlands
[4]Global Institute of Sustainability, Arizona State University, Tempe, AZ 85287, USA
[5]Research Domain Earth System Analysis, Potsdam Institute for Climate Impact Research,
Potsdam, Germany
[6]US Geological Survey, Earth Resources Observation and Science Centre, North Central Climate
Science Centre, Fort Collins, CO, USA
[7]Fenner School of Environment and Society, The Australian National University,
Canberra, Australia
[8]CSIRO Land and Water, Canberra, Australia
[9]Department of Atmospheric Science, Colorado State University, Fort Collins, USA

*Correspondence to:* L. Wang-Erlandsson (lan.wang@su.se)

**Abstract.** This study presents an "earth observation-based" method for estimating root zone storage capacity – a critical, yet uncertain parameter in hydrological and land surface modelling. By assuming that vegetation optimises its root zone storage capacity to bridge critical dry periods, we were able to use state-of-the-art satellite-based evaporation data computed with independent energy balance equations to derive gridded root zone storage capacity at global scale. This approach does not require soil or vegetation information, is model-independent, and is in principle scale-independent. In contrast to traditional look-up table approaches, our method captures the variability in root zone storage capacity within land cover types, including in rainforests where direct measurements of root depths otherwise are scarce. Implementing the estimated root zone storage capacity in the global hydrological model STEAM improved evaporation simulation overall, and in particular during the least evaporating months in sub-humid to humid regions with moderate to high seasonality. Our results suggest that several forest types are able to create a large storage to buffer for severe droughts (with very long return period), in contrast to for example savannahs and woody savannahs (medium length return period), as well as grasslands, shrublands, and croplands (very short return period). The presented method to estimate root zone storage capacity eliminates the need for poor resolution soil and rooting depth data that form a limitation for achieving progress in the global land surface modelling community.

# 1 Introduction

Root zone storage capacity ($S_R$) determines the maximum amount of soil moisture potentially available for vegetation transpiration, and is critical for correctly simulating deep drainage and surface runoff (Milly, 1994). Its parameterisation is also important for land-atmosphere interactions, the carbon cycle, and climate modelling (e.g., Bevan et al., 2014; Feddes et al., 2001; Hagemann and Kleidon, 1999; Hallgren and Pitman, 2000; Kleidon and Heimann, 1998b, 2000; Lee et al., 2005; Milly and Dunne, 1994; Zeng et al., 1998), and for irrigation management and crop yield models (e.g., Bastiaanssen et al., 2007; Hoogeveen et al., 2015).

However, root zone storage capacity is very difficult to measure and observe in the field, especially at the larger scales that are relevant for many modelling needs. Rooting profiles measurements are also scarce, and difficult to generalise since vegetation rooting systems naturally adapt to prevailing climates and soil heterogeneities (e.g., Gentine et al., 2012; Sivandran and Bras, 2013). Even when rooting profiles are available, difficulties arise in translating them to root zone storage capacity, due to variations in root densities, hydrological activity, horizontal spatial heterogeneities, and uncertainties in soil profile data including hard pans.

## 1.1 Background

Broadly six types of approaches to estimate the root zone storage capacity have been suggested or are in use in hydrological and land surface models: the *field observation based approach*, the *look-up table approach*, the *optimisation approach*, the *inverse modelling approach*, the *calibration approach*, and the *mass balance based approach*. These approaches are described below and compared in Table S1. Some of these approaches estimate rooting depth or root profiles, and can be translated to root zone storage capacity through combination with soil plant available water (Sect. 3.2., Eq. 11), even though it is a simplification.

The *field observation based approach* provide estimates of rooting depths based on rooting depth measurements (Doorenbos and Pruitt, 1977; Dunne and Willmott, 1996; Jackson et al., 1996; Schenk and Jackson, 2002; Zeng, 2001) and has the advantage of being constructed from actual observations of vertical rooting distribution (Canadell et al., 1996; Jackson et al., 1996). To scale up rooting depth to the global scale, Schenk and Jackson (2002) used the mean biome rooting depth and Schenk and Jackson (2009) employed an empirical regression model based on reported root profile from literature. However, this method suffers from data scarcity and location bias, and risks unlikely vegetation and soil combinations due to data uncertainty (Feddes et al., 2001). Moreover, it requires assumptions on water uptake from a certain fraction of the entire observed root profile. Observations show that many woody and herbaceous vegetation species are able to access very deep layers in a variety of soil conditions (Canadell et al., 1996; Stone and Kalisz, 1991), up to 18 m in Amazonian tropical forest (Nepstad et al., 1994), 53 m in the desert of the south-western United States (Phillips,

1963), and 68 m (possibly 140 m) in the central Kalahari dry savannah (Jennings, 1974). However, isolated roots that go very deep does not necessarily mean that vegetation across the landscape can
exploit the full soil to that depth.

The *look-up table approach* is used in hydrological and land surface modelling to parametrise root zone storage capacity based on literature values of mean biome rooting depth and soil texture data (e.g., Müller Schmied et al., 2014; Wang-Erlandsson et al., 2014). This approach facilitates land cover change experiments and is grounded in literature, but assumes root zone storage capacity to
be a function of merely land cover and soil type, with little consideration for climatic adjustments. This is a major oversight, as plants within the same vegetation type can exhibit a large span of root zone storage capacities in different climates and landscapes by adaptation to environmental conditions (Collins and Bras, 2007; Feldman, 1984; Gentine et al., 2012; Nepstad et al., 1994). Moreover, an incompatibility issue may arise if the literature based rooting depths employs a land
cover classification different from that of the land surface model (Zeng, 2001).

The *optimisation approach* predicts vertical rooting depth based on soil, climate, and vegetation data, and assumptions about the soil hydraulic properties and root distribution behaviour. Often, optimal root profiles are derived based on maximised net primary production (Kleidon and Heimann, 1998a), carbon or transpiration gain (e.g., Collins and Bras, 2007; Schwinning and Ehleringer, 2001;
van Wijk and Bouten, 2001), sometimes also while being as shallow as possible (e.g., Laio et al., 2006; Schenk, 2008). The optimisation techniques used differ widely, including genetic algorithm (Schwinning and Ehleringer, 2001; van Wijk and Bouten, 2001), physical ecohydrological modelling (Collins and Bras, 2007; Hildebrandt and Eltahir, 2007), simple analytical modelling (Laio et al., 2006), and stochastic modelling (Schenk, 2008). This approach is powerful for improving the
understanding of root profile development and can be useful for land surface models with explicit root distribution description (Smithwick et al., 2014). Nevertheless, further model development is needed to handle all types of environments (e.g., additional routines to handle groundwater uptake, acidic soil horizons, or low soil temperature) (Schenk, 2008).

The *inverse modelling approach* estimate rooting depth using a model to iteratively simulate a
variable available from satellite data (e.g., net or gross primary production, absorbed photosynthetically active radiation, or total terrestrial evaporation) with different rooting depth parameterisations (Ichii et al., 2007, 2009; Kleidon, 2004). This approach has a large spatial coverage while being indirectly observation-based, but is also dependent on soil information as well as the land surface model performance. Recently, this approach has also been applied at the local scale to approximate
the root zone storage capacity by minimising differences between evaporation modelled from water balance and evaporation from remote sensing (Campos et al., 2016).

The *calibration approach* is widely used in hydrology, whereby a hydrological model is calibrated on the root zone storage capacity, using hydrological records on precipitation, runoff and evaporation, sometimes in combination with expert knowledge (e.g., Feddes et al., 1993; Fenicia et al., 2008;

Jhorar et al., 2004; Winsemius et al., 2009; Gharari et al., 2014). However, the parameters derived are tied to the model used for calibration and are not necessarily comparable to measurable variables in nature, since they tend to compensate for uncertainties in model structure and data. In addition, since discharge is often the only observed variable (or one of only a few), the calibration approach is only suitable for applications at the catchment scale. For global hydrological models, parameters can be calibrated separately for a selection of gauged river basins and transferred to neighbouring ungauged catchments (Döll et al., 2003; Güntner, 2008; Hunger and Döll, 2008; Nijssen et al., 2001; Widén-Nilsson et al., 2007). This procedure, known as regionalisation, has (to our knowledge) only been performed for other parameter values than the root zone storage capacity, although the principle does not change with the parameters tuned. Nevertheless, challenges remain with discharge data uncertainty and parameter equifinality (Beven, 2006).

Recently, Gao et al. (2014) used a *mass balance approach* – more specifically, the mass curve technique – to estimate the root zone storage capacity at the catchment scale in the US and in Thailand. The underlying assumption is based on the tested hypothesis that plants will not root deeper than necessary (Milly and Dunne, 1994; Milly, 1994; Schenk, 2008). The water demand during the dry season equaled a constant transpiration rate, which was obtained through a water balance approach together with a normalised difference vegetation index (NDVI). Their results suggested that ecosystems develop their root zone storage capacity to deal with droughts with specific return periods, beyond which the costs of carbon allocation to roots are too high from the perspective of the plants. This resonates well with past economic analyses of plant behaviour and traits, e.g. Givnish (2014). Yet another mass balance approach was applied by de Boer-Euser et al. (2016) to catchments in New Zealand, using an interception and a root zone storage reservoir to record soil moisture storage deficit from variations in precipitation and transpiration. They derived mean annual transpiration from annual water balances, and seasonality of transpiration was added through estimate of potential transpiration and assumption about vegetation dormancy. The largest storage deficit of individual years were then used to derive catchment representative root zone storage capacity from Gumbel extreme value distribution assuming dry spell return periods of 10 years. These two applications of the mass balance approach have the advantage of being both model-independent and indirectly observation-based. In addition, no land cover or soil information is needed, making the method parsimonious and flexible. Irrigation was, however, not considered and their assumption of ecosystem adaptation does not apply very well to seasonal crops (de Boer-Euser et al., 2016).

In a similar cumulative mass balance approach, van Dijk et al. (2014) combined a satellite evapotranspiration product with monthly precipitation data to estimate a 'mean seasonal storage range' (MSSR) at 250 m resolution across Australia, as one of the inputs into national-scale mapping of groundwater dependent ecosystems (http://www.bom.gov.au/water/groundwater/gde/). MSSR expresses the estimated mean seasonal range in the amount of water stored in all water stores combined (surface, soil and groundwater). A large range was considered likely to indicate a large use of water

from storage during low rainfall periods from, for example, root water uptake from deeper soil or groundwater storages. Separate mapping of areas subject to irrigation or flood inundation was used to identify areas likely to rely on groundwater. The main conceptual drawback of this method is that the longer-term average seasonal pattern is likely to underestimate rooting depth in general, and even more so in regions without a strong seasonality in rainfall. The method also proved sensitive to any bias in evaporation and rainfall estimates and, in some conditions, simplifying assumptions about runoff and drainage rates (van Dijk et al., 2014).

## 1.2 Research aims

This study constitutes a first attempt to estimate global root zone storage capacity from satellite based evaporation and precipitation data using a mass balance approach, which is possible thanks to recent development, testing and validation of remote sensing evaporation products (e.g., Anderson et al., 2011; Guerschman et al., 2009; Hofste, 2014; Hu and Jia, 2015; Mu et al., 2011). Similar to the other mass balance based approaches, we assume that all hydrologically active roots are being used during the driest time and is not deeper than necessary. While we make use of the same mass balance principle as applied by Gao et al. (2014) and de Boer-Euser et al. (2016), our algorithm is based on indirect measurements of every unique pixel. Methodologically, in contrast to these two studies, the analyses here are carried out on global gridded data rather than by catchment and use total evaporation instead of interception and transpiration estimates.

Our aims are to: (1) present a method for estimating root zone storage capacity using remote sensing evaporation and precipitation data at global scale that includes the influence of irrigation; (2) evaluate how the new method influences evaporation simulation in a global hydrological model, in comparison to a classical look-up table approach; and (3) investigate the drought return periods different land cover types adjust to. This study, thus, provides an earth observation-based and model-independent estimate of global root zone storage capacity that can be useful in models without the need for root distribution and soil information.

## 2 Methods

### 2.1 Estimating root zone storage capacity

The root zone storage capacity $S_R$ is estimated from soil moisture deficit $D$ constructed from time series of water outflow $F_{out}$ and inflow $F_{in}$ from the root zone storage system. The algorithm is explained in this section and conceptually illustrated in Fig. 1.

First, we define the inflows and outflows from the system. The drying $F_{out}$ of the system is the total daily evaporation $E$:

$$F_{out} = E. \tag{1}$$

Note that the total evaporation $E$ is defined as the sum of transpiration, interception evaporation, soil moisture evaporation and open water evaporation.

The wetting $F_{\text{in}}$ of the system is the total daily precipitation $P$ and the effective irrigation water $F_{\text{irr}}$ (i.e., additional evaporation from surface, wet soil, and ponding water at the tail end of irrigation borders that originates from irrigation):

$$F_{\text{in}} = P + F_{\text{irr}}. \tag{2}$$

We need the term $F_{\text{irr}}$ in order to prevent $S_{\text{R}}$ from becoming overestimated in irrigated regions. This is because irrigation is captured in satellite-based evaporation data, but obviously not in precipitation data. Without correction, the irrigation evaporation in the satellite evaporation data would erroneously contribute to accumulation of soil moisture deficit in our computations. Beside irriga-
tion, additional evaporation from natural non-soil water storages (e.g, floodplains, wetlands, and groundwater) may contribute to overestimation of soil storage dynamics (see also Sect. 4.5 Limitations). In regions (see Appendix A) where the annual accumulated evaporation exceeds annual accumulated precipitation, also the long term average of the difference of $E - (P + F_{\text{irr}})$ is added to $F_{\text{in}}$ in order to compensate for lateral inflow or estimation errors in evaporation or precipitation.
Second, the difference between inflow and outflow is calculated at the daily scale. The accumulated difference $A$ is represented by the shaded areas in Fig. 1 and can be defined as

$$\underset{t_n \to t_{n+1}}{A} = \int\limits_{t_n}^{t_{n+1}} F_{\text{out}} - F_{\text{in}} \mathrm{d}t, \tag{3}$$

where $t_n$ is either the start of the accounting period or a point in time when $F_{\text{out}} = F_{\text{in}}$.

Third, we calculate the moisture deficit $D$, being the shortage of water from rainfall:

$$D(t_{n+1}) = \max\left(0, D(t_n) + \underset{t_n \to t_{n+1}}{A}\right). \tag{4}$$

The accumulation of $D$ will occur in our algorithm only during periods where $F_{\text{out}} > F_{\text{in}}$, and reductions of $D$ will occur when $F_{\text{out}} < F_{\text{in}}$. However, $D$ never becomes negative by definition, since it can be considered a running estimate of the root zone storage reservoir size (see Fig. 1 at $t_2$). Not allowing negative $D$ also means that any excess precipitation is assumed to be runoff or deep
drainage. In this way, for every hydrological year, one maximum accumulated moisture deficit can be determined, representing the largest annual drought. A long time series of these maximum annual values creates the opportunity to study the return period of the maximum moisture deficits. Extreme values analysis, such as by Gumbel's method (Gumbel, 1935), then yield estimates of extreme moisture deficits with different probabilities of exceedance, see Sect. 2.3.
Finally, the root zone storage capacity ($S_{\text{R}}$) is defined as the maximum of the obtained $D$ values:

$$\underset{t_0 \to t_{\text{end}}}{S_{\text{R}}} = \max\left(D(t_0), D(t_1), D(t_2), ..., D(t_{\text{end}})\right). \tag{5}$$

$S_\mathrm{R}$ estimate based on an evaporation and precipitation time series would (in the absence of additional water supply) theoretically constitute a *minimum* root zone storage capacity, see Fig. S1. If the root water uptake by plants does not abstract water until wilting point, the root zone storage may not utilise its full capacity. Note also that the $S_\mathrm{R}$ computed is not to be confused with time variable moisture availability. The time-variable water availability can be inferred from hydrological models using $S_\mathrm{R}$ as the water holding capacity.

During dry periods, the magnitude of surface runoff and deep drainage is usually small, and therefore is assumed to not affect root zone storage capacity calculations.

## 2.2 Implementation in a hydrological model

The newly derived root zone storage capacity is used in the global hydrological model STEAM (Wang-Erlandsson et al., 2014) to evaluate its influence on evaporation simulation. STEAM is a process-based model that partitions evaporation into five fluxes (i.e., vegetation interception, floor interception, transpiration, soil moisture evaporation, and open water evaporation). Potential evaporation is computed using the Penman-Monteith equation (Monteith, 1965), surface stomatal resistance is based on the Jarvis-Stewart equation (Stewart, 1988), and phenology is expressed as a function of minimum temperature, soil moisture content and daylight (Jolly et al., 2005). The model operates at $1.5°$ and 3 hours resolution.

In STEAM, root zone storage capacity is originally calculated as the product of soil plant available water (depending on soil texture) and rooting depth (depending on land cover type), using volumetric soil moisture as input to the stress function (here, the formulation of van Genuchten (1980)):

$$f(\theta) = \frac{\theta - \theta_\mathrm{wp}}{\theta_\mathrm{fc} - \theta_\mathrm{wp}}, \tag{6}$$

where $\theta$ is the actual volumetric soil moisture content (dimensionless), $\theta_\mathrm{wp}$ is the volumetric soil moisture content at wilting point, $\theta_\mathrm{fc}$ at field capacity. (This soil moisture stress function departs from the original formulation in STEAM (Matsumoto et al., 2008; Wang-Erlandsson et al., 2014), which is described in Sect. 2 in the Supplementary Information.)

However, the root zone storage capacity $S_\mathrm{R}$ is simply location-bound (depending on climatic variables alone) and no longer considered a land cover and soil based parameter. Thus, to use $S_\mathrm{R}$ directly, we do not account for soil moisture below wilting point and assume $S_\mathrm{R} = h(\theta_\mathrm{fc} - \theta_\mathrm{wp})$, where $h$ is the rooting depth (m). The reformulated stress function of soil moisture becomes:

$$f(S) = \frac{S}{S_R}, \tag{7}$$

where $S$ is the actual root zone storage (m). This reformulation is possible since the stress function retains its shape. Thus, $S_\mathrm{R}$ can in similar ways be implemented in other hydrological models.

To measure improvement, the root mean square error ($\varepsilon_\mathrm{RMS}$) for simulated evaporation is calculated using the original look-up table based root zone storage capacity $S_\mathrm{R,STEAM}$ and the newly

derived root zone storage capacity $S_{R,new}$ (i.e., $S_{R,CRU-SM}$ or $S_{R,CHIRPS-CSM}$) respectively. The root mean square error improvement ($\varepsilon_{RMS, imp}$) is positive if the $E$ simulated using $S_R$ is closer to a benchmark evaporation data set than the $E$ simulated using $S_{R,STEAM}$. The equation below shows the $\varepsilon_{RMS,imp}$ of $S_{R,new}$:

$$\varepsilon_{RMS, imp} = \varepsilon_{RMS}(E_{S_{R,STEAM}}, E_{benchmark}) - \varepsilon_{RMS}(E_{S_{R,new}}, E_{benchmark}).$$

The remote sensing based ensemble evaporation product $E_{SM}$ (and $E_{CSM}$, see Fig. S7 in the Supplementary Information) was used as benchmark $E_{benchmark}$. This use may seem circular when $E_{benchmark}$ is used to derive $S_{R,new}$, but is in fact valid due to differences in algorithms, precipitation input data, model types, and time span covered. First, the algorithms for estimating $S_{R,new}$, and for estimating $E$ in STEAM are very different. While $S_{R,new}$ is derived based on the $E$ overshoot over $P$, STEAM is a process-based model where evaporation originates from five different compartments, each constrained by potential evaporation and related stress functions. This means that it is impossible to reproduce $E_{benchmark}$ simply by inserting $S_{R,new}$ to STEAM. Second, the precipitation products (CRU and CHIRPS respectively) used for deriving $S_{R,new}$ differ from the precipitation forcing (ERA-I) used in STEAM. Third, $E_{benchmark}$ and STEAM are truly independent to each other as well. Whereas STEAM is process and water balance based, the ensemble $E$ product is based on a combination of two ($E_{SM}$) or three ($E_{CSM}$) energy balance methods. Last, $S_{R,new}$ is based on a single year value of $E_{benchmark}$ (i.e., the year of maximum storage deficit), whereas the analyses of improvements are based on the entire available time series of 10-11 years. The only difference of the new STEAM simulations is the inclusion of updated information on root zone storage so that during longer periods of drought, more realistic estimations of continued evaporation processes can be expected. Thus, if $S_{R,new}$ dimensioned on one year of $E_{benchmark}$ nevertheless improves $E$ simulation in STEAM with regard to 10-11 years of $E_{benchmark}$ (i.e., the overall $\epsilon_{RMS}$ decreases when $S_{R,new}$ is used in STEAM) is a strong indication that the storage capacity correction was implemented for the right reason.

To investigate where the performance increases are most significant, improvements in mean annual, mean maximum monthly and mean minimum monthly $E$ is calculated separately. $\varepsilon_{RMS,imp}$ by climate are done for bins of precipitation seasonality index and aridity index (defined in Appendix B) containing more than 200 grid cells. $\varepsilon_{RMS,imp}$ by land cover types are analysed for grid cells where single land cover occupancy exceeds 90 % in a 1.5° grid cell. $\varepsilon_{RMS}$ analyses are carried out on area weighted evaporation values to avoid bias caused by differences in grid cell areas. Results are shown in Sect. 4.4.

## 2.3 Frequency analysis

We calculate $S_R$ for 10 to 11 years (2003–2012 and 2003–2013 respectively, see Sect. 3.1) depending on data availability. However, different ecosystems may adapt their root system depths to different

return periods of drought which may or may not correspond to the available data time series length. Thus, we also determine the $S_{\text{R},L\text{ yrs}}$ for different return periods of drought $L$ (see Sect. 4.4) based on Gumbel's distribution (Gumbel, 1935). The resulting $S_{\text{R},L\text{ yrs}}$ is a function of the mean and standard deviation of the extremes in the data series:

$$S_{\text{R},L\text{ yrs}} = \overline{S_{\text{R}}} + \frac{\sigma_{S_{\text{R}}}}{\sigma_n}(y_L - y_n), \tag{8}$$

where $y_n$ is the reduced mean as a function of the number of available years $n$ ($y_{10} = 0.4952$ and $y_{11} = 0.4996$), $\sigma_n$ is the reduced standard deviation as a function of $n$ ($\sigma_{10} = 0.9496$ and $\sigma_{11} = 0.9676$), $\sigma_{S_{\text{R}}}$ is the standard deviation of $S_{\text{R}}$, while $y_L$ is the reduced variate of the Gumbel distribution:

$$y_L = -\ln\left(-\ln\left[1 - \frac{1}{L}\right]\right). \tag{9}$$

## 3 Data

### 3.1 Evaporation and precipitation input for estimating $S_{\text{R}}$

We present two $S_{\text{R}}$ datasets, one covering the latitudes 50° N–50° S ($S_{\text{R,CHIRPS-CSM}}$), and one with global coverage 80° N–56° S ($S_{\text{R,CRU-SM}}$). See Table 1 for an overview of the data input for each $S_{\text{R}}$ dataset.

For the clipped 50° N–50° S $S_{\text{R,CHIRPS-CSM}}$ map, we matched the 0.05° USGS Climate Hazards Group InfraRed Precipitation with Stations (CHIRPS) precipitation data ($P_{\text{CHIRPS}}$) (Funk et al., 2014) with the ensemble mean of three satellite-based global scale evaporation datasets ($E_{\text{CSM}}$): the CSIRO MODIS Reflectance Scaling EvapoTranspiration (CMRSET) v1405 at 0.05° (Guerschman et al., 2009), the Operational Simplified Surface Energy Balance (SSEBop) at 30″ (Senay et al., 2013), and the MODIS evapotranspiration (MOD16) at 0.05° (Mu et al., 2011). These three different evaporation models are all based on MODIS satellite data, but they use different parts of the electro-magnetic spectrum. CMRSET combines a vegetation index, which estimates vegetation photosynthetic activity, and shortwave infrared spectral data to estimate vegetation water content and presence of standing water. SSEBop relies on the thermal infrared data for determination of the latent heat flux and MOD16 on the visible and near-infrared data to account for Leaf Area Index variability. Hence, their input data, model structure and output data are not necessarily similar, which makes them attractive for deriving an ensemble evaporation product. $S_{\text{R,CHIRPS-CSM}}$ is based on data covering the years 2003–2012 as CMRSET was not available for 2013.

For the global coverage $S_{\text{R,CRU-SM}}$ map, we used the 0.5° Climatic Research Unit Timeseries version 3.22 (CRU TS3.22) precipitation data ($P_{\text{CRU}}$) (Harris et al., 2014) together with the ensemble mean ($E_{\text{SM}}$) of only SSEBop and MOD16, since we found CMRSET to overestimate evaporation at high latitudes, possibly due to the effect of snow cover on estimates. In addition, the irrigation effect was analysed for $S_{\text{R,CRU-SM}}$ by including evaporation originating from irrigation water simulated at 0.5° and at the daily scale by the dynamic global vegetation model LPJmL (Jägermeyr

et al., 2015). $S_{\mathrm{R,CRU-SM}}$ is computed based on evaporation data covering the years 2003–2013. Irrigation data cover the years 2003–2009 (monthly mean irrigation evaporation were used for years after 2009).

We present $S_{\mathrm{R,CHIRPS-CSM}}$, because $P_{\mathrm{CHIRPS}}$ is the lead precipitation product and we can make use of three evaporation datasets. However, $P_{\mathrm{CHIRPS}}$ is unfortunately not available at the global
scale, and CMRSET is not reliable in high latitudes. Thus, we added the global scale $S_{\mathrm{R,CRU-SM}}$ to this study. This allows for application in global scale models as well as investigations at the global scale (e.g., climate and land cover based analyses).

The input precipitation and evaporation data are shown in Figs. 2 and S2. This study required global coverage data at a grid cell resolution for both evaporation and precipitation. Importantly,
these products must not be produced using assumptions on root zone storage capacity, to prevent circularity (since we are estimating root zone storage capacity). In other words, there should be no water balance type of computation process involved in the determination of $S_{\mathrm{R}}$. We used satellite-based evaporation products because they are the only options available that fulfill these criteria, (i.e., reanalyses and land surface model evaporation contain soil depth information, whereas FLUXNET data
are too sparse for acquiring consistently good quality global coverage). The monthly satellite-based evaporation data used in the manuscript were those available at the time of this research. Conversely, precipitation data do not need to be satellite-based, but can also be ground-based. Inter-comparisons of precipitation products show that both CRU and CHIRPS are good quality precipitation products. In particular, CHIRPS performance stands out in a comprehensive inter-comparison of 13 difference
precipitation products in the Nile basin (Hessels, 2015). Nevertheless, data uncertainties still persist. The mean annual accumulated evaporation of $E_{\mathrm{CSM}}$ and $E_{\mathrm{SM}}$ is sometimes higher than the mean annual accumulated precipitation $P_{\mathrm{CHIRPS}}$ and $P_{\mathrm{CRU}}$, which is discussed in Appendix A. The use of three evaporation datasets decreases uncertainties related to individual evaporation products, because there is simply not one single preferred model. To compare the effect of different input data,
we also present results of $S_{\mathrm{R}}$ based on the separate evaporation and precipitation data (Figs. S4 and S5).

In addition, ECMWF re-analysis interim (ERA-I) (Dee et al., 2011) daily $0.5°$ evaporation and precipitation data were used to temporally downscale the monthly evaporation and precipitation data. In the temporal downscaling, we first established the ratios between daily values to the mean
monthly ERA-I, and second, used the relationship to estimate daily values from monthly $E_{\mathrm{SM}}$ or $E_{\mathrm{CSM}}$ values. This allows daily products of evaporation and precipitation, which was necessary in order to incorporate also short drought periods.

### 3.2 Other data used in analyses

The following datasets were compared with our $S_{\mathrm{R}}$ estimates:

– the estimated $1°$ rooting depth for 95 % of the roots from Schenk and Jackson (2009);

- the 1° rooting depth estimated by the optimised inverse modelling from Kleidon (2004), (where the minimum rooting depth producing the long-term maximum net primary production is selected as the best estimate);

- the 1° rooting depth estimated by the assimilated inverse modelling from Kleidon (2004), (where the rooting depth that minimises the difference between the modelled and the satellite-derived absorbed photosynthetically active radiation is selected as the best estimate); and

- the root zone storage capacity look-up table based parametrisation used in a global hydrological model, i.e., the Simple Terrestrial Evaporation to Atmosphere Model (STEAM)(Wang-Erlandsson et al., 2014).

In order to enable comparison between rooting depth $h$ and root zone storage capacity $S_\mathrm{R}$, we assumed that the root zone reaches its wilting point and converted between $h$ and $S_\mathrm{R}$ using soil properties:

$$S_\mathrm{R} = h\theta_\mathrm{paw} = h\left(\theta_\mathrm{fc} - \theta_\mathrm{wp}\right), \tag{10}$$

where $\theta_\mathrm{paw}$ is the maximum plant available soil moisture, $\theta_\mathrm{fc}$ is the volumetric soil moisture content at field capacity and $\theta_\mathrm{wp}$ is the volumetric soil moisture content at wilting point. Soil texture data at $30''$ is taken from the Harmonised World Soil Database (HWSD) (FAO/IIASA/ISRIC/ISSCAS/JRC, 2012), and field capacity and wilting point information is based on the United States Department of Agriculture (USDA) soil classification (Saxton and Rawls, 2006).

To analyse if and how the inferred $S_\mathrm{R}$ may improve simulations in a hydrological model, we applied $S_\mathrm{R,CRU-SM}$ to the evaporation simulation model STEAM. Input ERA-I data to STEAM were at 3 h and 1.5° resolution and include: precipitation, snowfall, snowmelt, temperature at 2 m height, dew point temperature at 2 m height, wind speed vector fields (zonal and meridional components) at 10 m height, incoming shortwave radiation, net long-wave radiation, and evaporation (only used to scale potential evaporation from daily to 3 h). To analyse the improvements in simulated evaporation by using $S_\mathrm{R,CRU-SM}$ as input to STEAM (see Sect. 4.3), we used an aridity index based on precipitation and reference evaporation from CRU TS3.22 (Harris et al., 2014).

For land cover-based analyses, we used the 0.05° Land Cover Type Climate Modeling Grid (CMG) MCD12C1 created from Terra and Aqua Moderate Resolution Imaging Spectroradiometer (MODIS) data (Friedl et al., 2010) for the year 2008, based on the land cover classification according to the International Geosphere – Biosphere Programme (IGBP) (shown in Fig. S3). Land cover fractions are preserved in upscaling to 0.5°. Only 0.5° grid cells containing at least 95 % of a single land cover type are used in the land cover-based analyses (see Sect. 4.2.2) and grid cells containing at least 95 % water are removed from all $S_\mathrm{R}$ analyses.

Data with finer resolution than 0.5° have been upscaled to 0.5° by simple averaging (i.e., assuming that the value of a 0.5° grid cell correspond to the mean of the overlapping finer grid cell values).

Data with coarser resolution than 0.5° were downscaled by oversampling (i.e., transferring grid cell values assuming that the finer 0.5° grid cell values correspond to those overlapped by the coarser degree grid cell values).

## 4 Results and discussion

### 4.1 Root zone storage capacity estimates

Figure 3 shows the $S_{\mathrm{R,CHIRPS-CSM}}$ (clipped, based on $E_{\mathrm{CSM}}$ and $P_{\mathrm{CHIRPS}}$) and $S_{\mathrm{R,CRU-SM}}$ (global, based on $E_{\mathrm{SM}}$ and $P_{\mathrm{CRU}}$) estimates adjusted for irrigation, (provided in the Supplements as ASCII-files). Independent of the input data used, large root zone storage capacities are observed in the semi-arid Sahel, South American and African savannah, central US, India, parts of Southeast Asia, and northern Australia. The lowest root zone storage capacities are observed in the most arid and barren areas, and in the humid and densely-vegetated tropics. The largest differences between $S_{\mathrm{R,CHIRPS-CSM}}$ and $S_{\mathrm{R,CRU-SM}}$ are observed over the Amazon, along the Andes, in Central Asia, and in the Sahara. Along mountain ridges (for example along the Andes and Himalaya), the $S_{\mathrm{R}}$ estimates are generally large, possibly due to data uncertainty in these transition regions or evaporation in foothills sustained by lateral water fluxes from the mountains in addition to precipitation. The positive values of $S_{\mathrm{R,CHIRPS-CSM}}$ in the Sahara desert are caused by overestimation of evaporation in the CMRSET evaporation product, (see also Figs. S4 and S5).

Notably, Fig. 3 show contrasting root zone storage capacity over the South American and African tropical forests, although they belong to the same ecological class (i.e., evergreen broadleaf forests). This variability is purely due to temporal fluctuations between precipitation and evaporation and is independent of soil properties.

### 4.2 Comparison to other root zone storage capacity estimates

#### 4.2.1 Geographic comparison

Figure 4 shows root zone storage capacity estimates (directly determined or converted from rooting depth, see Sect. 3.2) from other studies and compares them to $S_{\mathrm{R,CRU-SM}}$. The estimates shown are based on: rooting depths containing 95 % of all roots from Schenk and Jackson (2009) ($S_{\mathrm{R,Schenk}}$, Fig. 4a), hydrologically active rooting depth from inverse modelling (Kleidon, 2004) using the optimisation ($S_{\mathrm{R,Kleidon,O}}$, Fig. 4c) and assimilation approach ($S_{\mathrm{R,Kleidon,A}}$, Fig. 4e), and from a literature-based look-up table used in the hydrological model STEAM ($S_{\mathrm{R,STEAM}}$, Fig. 4g) (Wang-Erlandsson et al., 2014).

When the different datasets are compared to $S_{\mathrm{R,CRU-SM}}$ (Fig. 4b, 4d, 4f and 4h), we see both agreements and significant differences. All datasets appear to more or less agree on the approximate range of root zone storage capacity in large parts of the Northern Hemisphere. Around the Equator,

all datasets indicate root zone storage capacity to be lower or similar to that of $S_{\mathrm{R,CRU-SM}}$ in the tropical forests of the Amazon and the Indonesian islands. In the Congo region and in Central America, $S_{\mathrm{R,Kleidon,O}}$ and $S_{\mathrm{R,Kleidon,A}}$ are larger than both $S_{\mathrm{R,CRU-SM}}$ and the other. In the south temperate zone, $S_{\mathrm{R,CRU-SM}}$ appear to correspond to or be lower than the other datasets.

Figure 4 also reveal patterns specific to the different datasets that can be explained by the underlying method used for estimating rooting depth or root zone storage. For example, both $S_{\mathrm{R,Schenk}}$ and $S_{\mathrm{R,STEAM}}$ contain spuriously large values in the deserts (such as the Sahara and the Gobi) where vegetation is non-existent or extremely sparse. The methods based on satellite data ($S_{\mathrm{R,CRU-SM}}$, $S_{\mathrm{R,Kleidon,O}}$ and $S_{\mathrm{R,Kleidon,A}}$) appear to reflect reality in deserts more accurately. The $S_{\mathrm{R,Kleidon,O}}$ presents the largest root zone storage capacities (most pronounced over Africa, India, parts of South America), since this dataset represent an idealised and optimised case. On the contrary, the smallest root zone storage capacities are presented in the Amazon rainforest by $S_{\mathrm{R,Schenk}}$. These smaller values could be due to the lack of observations, since $S_{\mathrm{R,Schenk}}$ is derived from rooting depth field measurements. But any difference between rooting depth and root zone storage capacity could also be due to discrepancies between actual rooting depth and hydrologically active rooting depth (see also Sect. 3.2). In contrast to the other datasets, $S_{\mathrm{R,STEAM}}$ is relatively homogenous and does not contain any large values (basically all $< 400\,\mathrm{mm}$) (Fig. 4g). This is natural, since the other datasets are based on more heterogeneous observations, whereas $S_{\mathrm{R,STEAM}}$ is based on a homogenous look-up table. Nevertheless, different input data were also used in the different studies. Thus, it is difficult to attribute the variations in root zone storage capacity estimates to differences in methods or differences in input data. Additional comparisons in scatter plots and root mean square error are shown in Fig. S6 and Table S2.

### 4.2.2 Distribution by land cover type

Figure 5 shows the root zone storage capacity distribution for different land cover types and $S_{\mathrm{R}}$ datasets, ($S_{\mathrm{R,CHIRPS-CSM}}$ is not shown since it does not have global coverage). Except for deciduous broadleaf forests, the $S_{\mathrm{R,CRU-SM}}$ of forests (Fig. 5a–e) are closer to $S_{\mathrm{R,Kleidon,O}}$ and $S_{\mathrm{R,Kleidon,A}}$ than to $S_{\mathrm{R,Schenk}}$. Interestingly, the range of $S_{\mathrm{R}}$ is large in the evergreen forest types for the "adaptive" estimates ($S_{\mathrm{R,CRU-SM}}$, $S_{\mathrm{R,Kleidon,O}}$, and $S_{\mathrm{R,Kleidon,A}}$), but small for the literature based methods ($S_{\mathrm{R,Schenk}}$ and $S_{\mathrm{R,STEAM}}$). In open shrublands and grasslands (Fig. 5f and i) root zone storage capacities are similar across all estimates, except for the higher $S_{\mathrm{R,STEAM}}$. In savannahs, croplands, and natural/vegetation mosaic areas (Fig. 5h, j, k), $S_{\mathrm{R,Kleidon,O}}$, and $S_{\mathrm{R,Kleidon,A}}$ appear to have higher values than others. In woody savannahs (Fig. 5g), $S_{\mathrm{R,Kleidon,O}}$ has a notably large range as well as high mean root zone storage capacity. In barren land (Fig. 5l), $S_{\mathrm{R,Schenk}}$ and $S_{\mathrm{R,STEAM}}$ are counter-intuitively high.

### 4.3 Implementation in a hydrological model

We implemented $S_{\mathrm{R,CRU-SM}}$, $S_{\mathrm{R,CHIRPS-CSM}}$, and $S_{\mathrm{R,STEAM}}$ in the hydrological model STEAM
(see Sect. 2.2 for methods) in order to analyse how the new root zone storage capacities might
improve model performance. This section shows the performance analyses using $S_{\mathrm{R,CRU-SM}}$ as
input, since it has global coverage. A comparison in $E$ simulation performance between using
$S_{\mathrm{R,CHIRPS-CSM}}$ and $S_{\mathrm{R,CRU-SM}}$ as input to STEAM is shown in Fig. S7, and discussed in the
Supplementary Information.

Figure 6 compares the STEAM-simulated evaporation when using, on the one hand, $S_{\mathrm{R,CRU-SM}}$
and, on the other, the look-up table based $S_{\mathrm{R,STEAM}}$. In general, $S_{\mathrm{R,CRU-SM}}$ estimated higher evap-
oration rates in the tropics and lower evaporation in the subtropics and temperate zone. In particular,
the differences are pronounced during the warm and dry seasons. For example, the evaporation re-
ductions with $S_{\mathrm{R,CRU-SM}}$ is widespread in the Northern Hemisphere during July. During the dry
seasons (e.g., January in the Sahel, July in Congo south of the Equator), the evaporation increase is
the most significant. Moreover, the changes in evaporation also depend on land cover type. In South
America, evaporation increases in the seasonal tropical forests of the Amazon, whereas evaporation
decreases in the savannas and shrublands in the south. These results suggest that $S_{\mathrm{R,CRU-SM}}$ has
the greatest potential to influence model simulations for the hot and dry seasons, in regions where
the root zone storage varies strongly.

Figure 7 shows the $\varepsilon_{\mathrm{RMS}}$ improvements of simulated mean annual, mean maximum monthly and
mean minimum monthly $E$ sorted by seasonality and aridity, using $S_{\mathrm{R,CRU-SM}}$ as input and $E_{\mathrm{SM}}$
as benchmark. The analysis reveals that our $S_{\mathrm{R,CRU-SM}}$ estimate has the greatest potential to im-
prove model simulations for minimum monthly evaporation. In particular, the improvements become
significant with increased seasonality of rainfall, and in subhumid to humid regions, resonating the
findings of de Boer-Euser et al. (2016).

### 4.4 The effect of different drought return periods

Vegetation may adapt to a different time period than the 10–11 years of data that were available for
this study. Thus, we normalised $S_{\mathrm{R}}$ using the Gumbel distribution in order to assess the effect of
different drought return periods (see Sect. 2.3). Normalised $S_{\mathrm{R}}$ are provided in the Supplements as
ASCII-files.

Figure 8 shows the mean latitudinal $S_{\mathrm{R,CHIRPS-CSM},L\ \mathrm{yrs}}$ and $S_{\mathrm{R,CRU-SM},L\ \mathrm{yrs}}$ for differ-
ent drought return periods $L$ based on the Gumbel distribution. As may be expected, both
$S_{\mathrm{R,CHIRPS-CSM}}$ and $S_{\mathrm{R,CRU-SM}}$ based on the 10–11 years where data were available correspond
most closely to the $S_{\mathrm{R},L\ \mathrm{yrs}}$ for $L = 10$ years ($S_{\mathrm{R,10\ yrs}}$). $S_{\mathrm{R},L\ \mathrm{yrs}}$ always increases with $L$, but more
strongly for small $L$ and less so for large $L$ following the Gumbel distribution. The largest spans are
seen in the northern latitudes and around the equator.

Figure 9 shows a comparison of how Gumbel normalised $S_{R,CRU-SM,L \text{ yrs}}$ affect the evaporation simulation $\varepsilon_{RMS}$ improvements by land cover type. Interestingly, a drought return period of 2 years ($S_{R,CRU-SM,2 \text{ yrs}}$) offers the best evaporation simulation performance in deciduous broadleaf forests, open shrublands, grasslands, croplands and barren lands, whereas $S_{R,CRU-SM,10 \text{ yrs}}$ or $S_{R,CRU-SM,20 \text{ yrs}}$ are best in evergreen needleleaf forests, woody savannahs, and savannahs, and $S_{R,CRU-SM,60 \text{ yrs}}$ is best in evergreen broadleaf forests, deciduous needleleaf forests, and mixed forests.

A short drought return period of 2 years improves evaporation simulation the most in short vegetation types probably because these land cover types adapt to average years rather than to extreme drought years. In extreme years, they survive by going dormant. Evergreen broadleaf forests, on the other hand, adapt to 40–60 years of drought return period since they deal with droughts by accessing deeper soil moisture storages and thus invest in root growth (Brunner et al., 2015). The performance increases in deciduous needleleaf forests by using 60 years of drought return period could be explained by their need to cater for dry periods during their most active summer months. Shedding the leaves during the wet season (semi-arid tropics) or the growing season (summer in temperate climates) is not attractive because it prevents reproduction. Interestingly, deciduous broadleaf forests appear to adapt to a 2 years drought return period - i.e., radically different to deciduous needleleaf forests. This is possibly due to their younger age (Poulter, 2012; Hicke et al., 2007) and considerably shorter longevity (Larson, 2001; Loehle, 1988). Longevity could be explained by strong defence mechanisms against fungi and insects, lack of physical environmental damage, but also low occurrence of environmental stress such as drought (Larson, 2001). Thus, it seems logical that the older and longer living deciduous needleleaf forests have developed their root zone storage capacities to stand against more extreme droughts. Analysing the performance by each land cover type reveals interesting patterns (such as the contrast between deciduous needleleaf and broadleaf forests), but also leads to small sample sizes (particularly for evergreen needleleaf forests and the deciduous forest types) that should be considered when interpreting the results.

Based on the best performing drought return periods for each land cover types, we created a Gumbel normalised root zone storage capacity map (Fig. S9), which is shown and analysed in Sect. 3 in the Supplementary Information. In addition, we also analyse how $S_R$ of different land cover types can be associated with climatic indicators in Appendix B.

## 4.5 Limitations

Although research indicates that most ecosystem rooting depth are limited by water rather than other resources (Schenk, 2008), other factors may still cause $S_R$ to be larger than what is considered here. A minimum rooting depth of 0.3–0.4 m are for example considered in Schenk and Jackson (2009). Although we are comparing others' rooting depth estimates to $S_{R,CRU-SM}$, they are not directly comparable. Our approach deals with the accessible water volume in the root zone, which is not

always related to root zone depth since the root density can vary over the depth. Our $S_{\mathrm{R}}$ estimates implicitly capture the root density that is active in water uptake.

The $S_{\mathrm{R,CHIRPS-CSM}}$ and $S_{\mathrm{R,CRU-SM}}$ have been derived using evaporation and precipitation data from recent years (i.e., the 2000s), and should be used with caution if applied to past or future model simulations. Land cover change during the years 2003–2013 have not been taken into account. This has potential impact on the computation of additional evaporation from irrigated areas with fast changing acreages.

Wetlands and groundwater dependent ecosystems produce additional evaporation that cannot be ascribed to local rainfall (van Dijk et al., 2014). Bastiaanssen et al. (2014) recently demonstrated for the Nile basin that in some areas, natural withdrawals exceed man-made withdrawals to the irrigation sector. Since satellite evaporation data captures all types of evaporation, and we only corrected for irrigation, natural additional evaporation sources are implicitly included in $S_{\mathrm{R,CHIRPS-CSM}}$ and $S_{\mathrm{R,CRU-SM}}$. Thus, our $S_{\mathrm{R}}$ estimates may not strictly represent the root zone storage capacities in regions where water uptake from groundwater is significant, see Fig. A1.

The choice of remotely sensed evaporation products influenced the resulting $S_{\mathrm{R}}$ more than the choice of precipitation product in this study, see Fig. S4. In particular, the largest standard deviations in the ensemble evaporation products are located in central South America, the Sahel, India, and northern Australia (see Fig. 2e, 2f). To reduce uncertainty, the presented method is preferably applied using ensemble products based on reliable evaporation and precipitation datasets identified in comparison and evaluation studies (e.g., Hofste, 2014; Hu et al., 2015; Yilmaz et al., 2014; Trambauer et al., 2014; Bitew and Gebremichael, 2011; Herold et al., 2015; Hessels, 2015; Hu and Jia, 2015; Moazami et al., 2013; Trenberth et al., 1991).

Finally, while the $S_{\mathrm{R}}$ estimates are model independent, the analyses of the best performing drought return periods of different land cover types will depend on the hydrological model used, given the large variations of evaporation estimates (and in particular transpiration/evaporation ratios) among land surface models (e.g., Wang and Dickinson, 2012). Thus, although the contrasting return periods for woody land cover types and annual short vegetation types are supported by current knowledge about ecohydrological response to droughts, the calculated values are subject to assumptions. Uncertainties are probably largest for heterogeneous land cover types (such as savannahs) because they tend to be challenging to parameterise and simulate. Therefore, implementation of $S_{\mathrm{R}}$ in other hydrological or land surface models would require model-specific analyses of optimal return periods.

## 5   Summary and conclusion

This study presents a method to estimate root zone storage capacity in principle from remotely sensed evaporation and observation-based precipitation data, by assuming that plants do not invest more in their roots than necessary to bridge a dry period. Two global root zone storage estimates

($S_{\mathrm{R,CRU-SM}}$ and $S_{\mathrm{R,CHIRPS-CSM}}$) are presented based on different precipitation and evaporation datasets, but show in general similar patterns globally. $S_{\mathrm{R,CRU-SM}}$ and $S_{\mathrm{R,CHIRPS-CSM}}$ both improve mean annual $E$ simulation in STEAM (see Fig. S7), and there is not a preferred product.

Different ecosystems have evolved to survive droughts of different return periods with different strategies. Our analyses showed that whereas long drought return period increased performance for many forest types, short drought return period increased performance for many short vegetation types. The best $E$ simulation results were achieved when normalising the $S_{\mathrm{R}}$ estimate using a very short drought return period ($\sim$2 years) for deciduous broadleaf forests, grasslands, shrublands, croplands, and barren or sparsely vegetated lands, a medium length drought return period ($\sim$10-20 years) for evergreen needleleaf forests, woody savannahs, and savannahs, and a very long drought return period ($\sim$60 years) for evergreeen broadleaf, deciduous needleleaf, and mixed forests. This is probably because grasslands survive extreme droughts by going dormant, whereas forests invest in root growth (Brunner et al., 2015). Thus, the root zone storage capacities of short vegetation types seem to adapt to average years, whereas those of forests adapt to extreme years. Differences among forest types are thought to be related to forest age and drought coping strategy. Normalisation to longer drought return periods should not be done for short-lived annual plants such as two third of the world's croplands (Cox et al., 2006), nor beyond the age of the ecosystem of concern, because vegetation can not be assumed to adapt beyond their age.

The $S_{\mathrm{R}}$ estimates presented here are both globally gridded and observation-based. They have the advantage over the field study based and statistically derived $S_{\mathrm{R,Schenk}}$ (Schenk and Jackson, 2009) by being directly based on gridded data and by covering regions where observational studies are limited (e.g., the evergreen broadleaf forests). In comparison to the inverse modelling approaches of Kleidon (2004), the method presented in this study is independent of model simulations and therefore closer to direct observations.

The new $S_{\mathrm{R}}$ estimates can be used in hydrological and land surface modelling to improve simulation results, particularly in the dry season and in seasonal tropical forests where variations of root zone storage capacity are large. Using the new $S_{\mathrm{R}}$ as input to the hydrological model STEAM improved the evaporation simulation considerably in subhumid to humid regions with high seasonality. In particular, the most significant improvements occurred in the months with the least evaporation. Normalisation of $S_{\mathrm{R}}$ to different drought return periods for different land cover types could further improved evaporation simulation in STEAM, suggesting that Gumbel normalisation is a viable method to optimise the $S_{\mathrm{R}}$ estimates prior to implementation in global hydrological or land surface models.

The presented method is simple to apply and in principle scale-independent. For researchers working at regional or local scales, root zone storage capacities can easily be derived using available evaporation and precipitation data. Moreover, when information on irrigation and groundwater use is available, they can be used to adjust $S_{\mathrm{R}}$, as was done by for example van Dijk et al. (2014).

Satellite-based evaporation datasets are also quickly being developed and improved. New global scale evaporation products such as ALEXI (Atmosphere-Land Exchange Inverse) (Anderson et al., 2011) and ETMonitor (Hu and Jia, 2015) are underway based on 375 and $1000\,\mathrm{m}$ pixels. More sophisticated two-layer surface energy balance models also have the capacity to distinguish transpiration from other forms of evaporation. This implies that local root zone storage capacity can be computed, based on transpiration fluxes, which is preferred from a bio-physical point of view (although it would require estimate of interception evaporation to calculate effective precipitation). As new evaporation datasets become available, the $S_R$ estimates can easily be updated. In addition, this method can be used to diagnose and compare different evaporation products, in particular for identifying variations in seasonality. With longer time series of land cover and climate data, this method can possibly also be used to infer the effect of climate change on root zone storage capacity as a function of the adaptability of vegetation to altered conditions.

## Appendix A: Evaporation exceedance over precipitation

The mean annual accumulated evaporation of $E_{\mathrm{CSM}}$ and $E_{\mathrm{SM}}$ is sometimes higher than the mean annual accumulated precipitation $P_{\mathrm{CHIRPS}}$ and $P_{\mathrm{CRU}}$ (see Fig. A1). In these areas, overestimation of $S_R$ may be expected, because it is unlikely that the 10 or 11 year accumulation of $E$ is more than rainfall, except for hydrological situations with lateral inflow through inundation, irrigation or groundwater inflow. The evaporation dataset $E_{\mathrm{CSM}}$ exhibits larger and more widely spread exceedance over $P_{\mathrm{CHIRPS}}$ in comparison to the $E_{\mathrm{SM}} - P_{\mathrm{CRU}}$ combination. Most notably, the exceedance is high and potentially spurious in arid and semi-arid zones (e.g., the Sahara, western US, and Central Asia) which suggests that the evaporation from deserts is not accurate. Regions where both $S_{\mathrm{R,CRU-SM}}$ and $S_{\mathrm{R,CHIRPS-CSM}}$ show high accumulated evaporation exceedance are along the Andes, patches in western US, East Africa, Ivory Coast, Central Asia, Northwest China and spots in Australia. These are essentially irrigated areas, lakes, reservoirs, wetlands and coastal deltas. Possibly, overestimation of $S_R$ can also be caused for example by vegetation tapping into groundwater. Uncertainty in evaporation and precipitation products also propagates to errors in $S_R$. The uncertainty of evaporation is location specific, (grid cells with a large standard deviation between the individual $E$ products are shown in Fig. 2e and f).

Interestingly, the high evaporation exceedance appears to be much more pronounced during drier years. In Fig. A2, we sort every grid cell by the annual precipitation amount, from dry to wet, and plot the mean latitudinal $E$ exceedance for the regions where the long term accumulated $E - P$ is positive. The figure clearly shows that $E$ exceedance decreases with increase in rainfall, indicating that increased water demand during dry years is satisfied by withdrawing moisture from the soil matrix that is bounded with more potential (higher pF), or from underlying groundwater through deeply rooting vegetation.

## Appendix B: Climatic influence on root zone storage capacity depending on land cover type

### B1    Methods and data

We analyse how $S_{\mathrm{R,CRU-SM}}$ of different land cover types can be associated with climatic indicators. Stepwise multiple regression method based on the Akaike information criterion (AIC) is used to analyse how these climatic indicators may explain variations in $S_{\mathrm{R}}$ within a land cover type. The climatic indicators used are precipitation seasonality ($I_{\mathrm{s}}$), aridity ($I_{\mathrm{a}}$), and interstorm duration ($I_{\mathrm{isd}}$) (as these were found to be important by Gao et al. (2014)):

$$I_{\mathrm{S}} = \frac{1}{\overline{P_{\mathrm{a}}}} \sum_{\mathrm{m=1}}^{m=12} \left| \overline{P_{\mathrm{m}}} - \frac{\overline{P_{\mathrm{a}}}}{12} \right|, \text{ and} \tag{B1}$$

$$I_{\mathrm{a}} = \frac{\overline{P_{\mathrm{a}}}}{\overline{E_{\mathrm{p}}}}, \tag{B2}$$

where $\overline{P_{\mathrm{m}}}$ is the mean precipitation of the month, $\overline{P_{\mathrm{a}}}$ is the mean annual precipitation, and $\overline{E_{\mathrm{p}}}$ is the potential evaporation. We defined $I_{\mathrm{isd}}$ as the mean continuous number of days per year without precipitation. Interaction effects between the variables are taken into account.

The climate variables interstorm duration, aridity and precipitation seasonality are developed based on monthly 0.5° reference evaporation from CRU TS3.22 (Harris et al., 2014) and monthly 0.5° precipitation for 1982–2009 from the Global Precipitation Climatology Centre (GPCC) (Schneider et al., 2011). Here, GPCC data (instead of CRU) are used in order to prevent false correlation with the CRU-based $S_{\mathrm{R,CRU-SM}}$.

### B2    Results and discussion

We use multiple linear regression to correlate $S_{\mathrm{R,CRU-SM}}$ values to climatic indicators, with the aim to investigate how well climate indicators can predict root zone storage capacities in different land cover types. It appears that climate indicators predict root zone storage capacities much better in evergreen forests than in short vegetation types. Figure B1 shows high $R^2$ in mostly evergreen forests; moderate $R^2$ in other forest types and croplands; and low $R^2$ in savannahs, shrublands and grasslands. This is probably because of their different drought survival strategies. While evergreen forests bridge droughts by water uptake from storage in their root zone, deciduous forests shed their leaves, and short vegetation types such as grasslands go dormant and decrease their transpiration to a minimum. The multiple linear regression model for $S_{\mathrm{R}}$ in croplands is moderately explained by climate indicators, potentially due to human management. All climate variables were selected by AIC in the multiple linear regression model (Table B1).

**The Supplement related to this article is available online at**
**doi:10.5194/hess-0-1-2016-supplement.**

*Acknowledgements.* This research was supported by funding from the Swedish Research Council (Vetenskapsrådet) and the Swedish Research Council Formas (Forskningsrådet Formas). The global evaporation data sets were made available by the USGS FEWS NET (part of the USGS EROS Centre) (SSEBop model) and the CSIRO (CMRSET model). Without these data sets, the global upscaling would not have been feasible. We are grateful to Ruud van der Ent, Ingo Fetzer, Tanja de Boer-Euser, Remko Nijzink, and Tim Hessels for valuable discussions during the manuscript preparation, and Axel Kleidon and Jochen Schenk for sharing and explaining their data. We also thank the two anonymous referees and Axel Kleidon, whose careful review helped improve and clarify this manuscript.

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

**Table 1.** Overview of the time period, latitudinal coverage and data input for the two root zone storage capacity $S_R$ datasets ($S_{R,CHIRPS-CSM}$ and $S_{R,CRU-SM}$) produced in this study.

| | $S_{R,CHIRPS-CSM}$ | $S_{R,CRU-SM}$ |
|---|---|---|
| Years | 2003–2012 | 2003–2013 |
| Latitude coverage | 50° N–50° S | 80° N–56° S |
| Monthly $P$ data input | CHIRPS | CRU |
| Monthly $E$ data input | Mean of CMRSET, SSEBop, and MOD16 ($E_{CSM}$) | Mean of SSEBop and MOD16 ($E_{SM}$) |
| Monthly irrigation data input | LPJmL (2003–2009) | LPJmL (2003–2009) |
| Daily $E$ and $P$ data for downscaling | ERA-I | ERA-I |

Trambauer, P., Dutra, E., Maskey, S., Werner, M., Pappenberger, F., van Beek, L. P. H., and Uhlenbrook, S.: Comparison of different evaporation estimates over the African continent, Hydrol. Earth Syst. Sci., 18, 193–212, doi:10.5194/hess-18-193-2014, 2014.

Trenberth, K. E., Dai, A., van der Schrier, G., Jones, P. D., Barichivich, J., Briffa, K. R., and Sheffield, J.: Global warming and changes in drought, Nat. Clim. Change, 4, 17–22, doi:10.1038/nclimate2067, 2013.

van Dijk, A., Warren, G., Van Niel, T., Byrne, G., Pollock, D., and Doody, T.: Derivation of data layers from medium resolution remote sensing to support mapping of groundwater dependent ecosystems, Tech. rep., A report for the National Water Commission, 27 pp., 2014.

van Genuchten, M. T.: A Closed-form Equation for Predicting the Hydraulic Conductivity of Unsaturated Soils, Soil Sci. Soc. of Am. J., 44, 892–898, doi:10.2136/sssaj1980.03615995004400050002x, 1980.

van Wijk, M. T. and Bouten, W.: Towards understanding tree root profiles: simulating hydrologically optimal strategies for root distribution, Hydrol. Earth Syst. Sci., 5, 629–644, doi:10.5194/hess-5-629-2001, 2001.

Wang, K. and Dickinson, R. E.: A Review of Global Terrestrial Evapotranspiration: Observation, Modeling, 860 Climatology, and Climatic Variability, Rev. Geophys., 50, RG2005, doi:doi:10.1029/2011RG000373, 2012.

Wang-Erlandsson, L., van der Ent, R. J., Gordon, L. J., and Savenije, H. H. G.: Contrasting roles of interception and transpiration in the hydrological cycle – Part 1: Temporal characteristics over land, Earth Syst. Dynam., 5, 441–469, doi:10.5194/esd-5-441-2014, 2014.

Widén-Nilsson, E., Halldin, S., and Xu, C-Y.: Global water-balance modelling with WASMOD-M: Parameter 865 estimation and regionalisation, J. Hydrol., 340, 105–118, doi:10.1016/j.jhydrol.2007.04.002, 2007.

Winsemius, H. C., Schaefli, B., Montanari, A., and Savenije, H. H. G.: On the calibration of hydrological models in ungauged basins: A framework for integrating hard and soft hydrological information, Water Resour. Res., 45, W12422, doi:10.1029/2009WR007706, 2009.

Yilmaz, M. T., Anderson, M. C., Zaitchik, B., Hain, C. R., Crow, W. T., Ozdogan, M., Chun, J. A., and Evans, 870 J.: Comparison of prognostic and diagnostic surface flux modeling approaches over the Nile River basin, Water Resour. Res., 50, 386–408, doi:10.1002/2013WR014194, 2014.

Zeng, X.: Global vegetation root distribution for land modeling, J. Hydrometeorol., 2, 525–530, doi:10.1175/1525-7541(2001)002<0525:GVRDFL>2.0.CO;2, 2001.

Zeng, X., Dai, Y.-J., Dickinson, R. E., and Shaikh, M.: The role of root distribution for climate simulation over 875 land, Geophys. Res. Lett., 25, 4533–4536, doi:10.1029/1998GL900216, 1998.

**Table A1.** Predictor variables selected by Akaike Information Criterion (AIC) for the different land cover types. The predictor variables are interstorm duration ($I_{\text{isd}}$), precipitation seasonality $I_{\text{s}}$, and aridity index ($I_{\text{a}}$).

| Land cover type | Predictor variables |
|---|---|
| 02:evergreen needleleaf forest | $S_{\text{R}} = I_{\text{isd}} + I_{\text{s}} + I_{\text{a}} + I_{\text{isd}}{:}I_{\text{a}} + I_{\text{s}}{:}I_{\text{a}}$ |
| 03:evergreen broadleaf forest | $S_{\text{R}} = I_{\text{isd}} + I_{\text{s}} + I_{\text{a}} + I_{\text{isd}}{:}I_{\text{s}} + I_{\text{s}}{:}I_{\text{a}}$ |
| 04:deciduous needleleaf forest | $S_{\text{R}} = I_{\text{isd}} + I_{\text{s}} + I_{\text{a}} + I_{\text{s}}{:}I_{\text{a}} + I_{\text{isd}}{:}I_{\text{a}} + I_{\text{isd}}{:}I_{\text{s}} + I_{\text{isd}}{:}\,I_{\text{s}}{:}I_{\text{a}}$ |
| 05:deciduous broadleaf forest | $S_{\text{R}} = I_{\text{isd}} + I_{\text{s}} + I_{\text{a}} + I_{\text{s}}{:}I_{\text{a}} + I_{\text{isd}}{:}I_{\text{s}}$ |
| 06:mixed forests | $S_{\text{R}} = I_{\text{isd}} + I_{\text{s}} + I_{\text{a}} + I_{\text{isd}}{:}I_{\text{a}} + I_{\text{isd}}{:}I_{\text{s}} + I_{\text{s}}{:}I_{\text{a}} + I_{\text{isd}}{:}I_{\text{s}}{:}I_{\text{a}}$ |
| 08:open shrublands | $S_{\text{R}} = I_{\text{isd}} + I_{\text{s}} + I_{\text{a}} + I_{\text{isd}}{:}I_{\text{a}} + I_{\text{isd}}{:}I_{\text{s}} + I_{\text{s}}{:}I_{\text{a}} + I_{\text{isd}}{:}I_{\text{s}}{:}I_{\text{a}}$ |
| 09:woody savannas | $S_{\text{R}} = I_{\text{isd}} + I_{\text{s}} + I_{\text{a}} + I_{\text{isd}}{:}I_{\text{a}} + I_{\text{s}}{:}I_{\text{a}}$ |
| 10:savannas | $S_{\text{R}} = I_{\text{isd}} + I_{\text{s}} + I_{\text{a}} + I_{\text{isd}}{:}I_{\text{a}} + I_{\text{s}}{:}I_{\text{a}} + I_{\text{isd}}{:}I_{\text{s}} + I_{\text{isd}}{:}I_{\text{s}}{:}I_{\text{a}}$ |
| 11:grasslands | $S_{\text{R}} = I_{\text{isd}} + I_{\text{s}} + I_{\text{a}} + I_{\text{isd}}{:}I_{\text{s}} + I_{\text{s}}{:}I_{\text{a}}$ |
| 13:croplands | $S_{\text{R}} = I_{\text{isd}} + I_{\text{s}} + I_{\text{a}} + I_{\text{isd}}{:}I_{\text{a}} + I_{\text{isd}}{:}I_{\text{s}} + I_{\text{s}}{:}I_{\text{a}} + I_{\text{isd}}{:}\,I_{\text{s}}{:}I_{\text{a}}$ |
| 15:cropland/natural veg. mosaic | $S_{\text{R}} = I_{\text{isd}} + I_{\text{s}} + I_{\text{a}} + I_{\text{isd}}{:}I_{\text{s}}$ |
| 17:barren or sparsely vegetated | $S_{\text{R}} = I_{\text{isd}} + I_{\text{s}} + I_{\text{a}} + I_{\text{s}}{:}I_{\text{a}} + I_{\text{isd}}{:}\,I_{\text{a}} + I_{\text{isd}}{:}I_{\text{s}}$ |

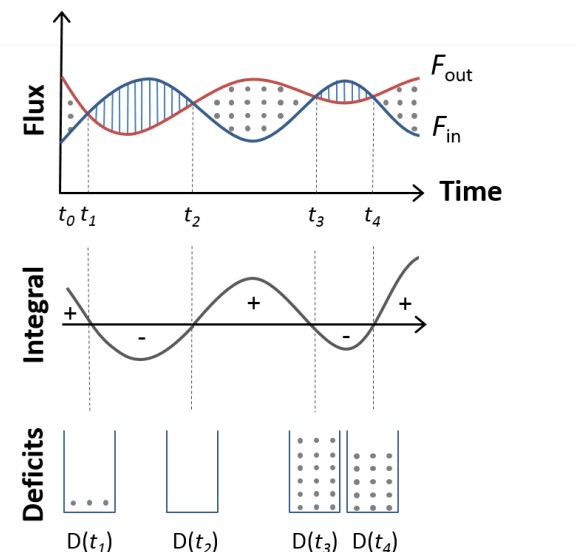

**Figure 1.** Conceptual illustration of the algorithm for calculating the root zone storage capacity $S_{\text{R}}$. The shaded areas represent the accumulated differences $A$ that are positive when outflow $F_{\text{out}} >$ inflow $F_{\text{in}}$, and negative when $F_{\text{out}} < F_{\text{in}}$. Moisture deficit $D$ is increased by positive $A$ and decreased by negative $A$. Note that $D$ never becomes negative.

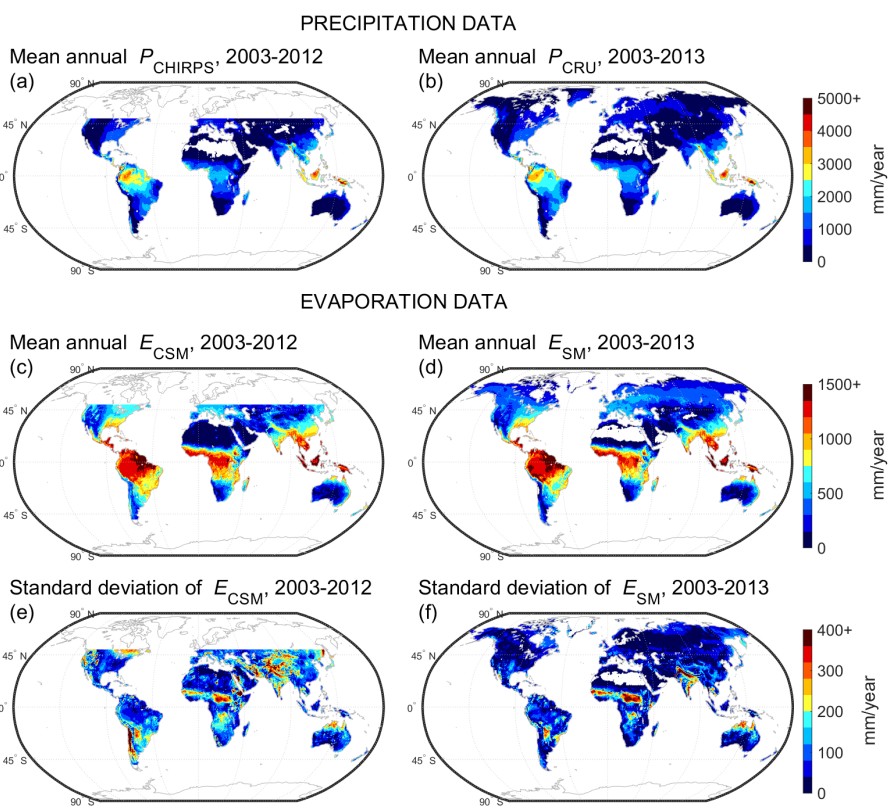

**Figure 2.** The mean annual precipitation of **(a)** CHIRPS ($P_{\mathrm{CHIRPS}}$) for the years 2003–2012 (50° N–50° S), and **(b)** CRU ($P_{\mathrm{CRU}}$) for the years 2003–2013 (80° N–56° S). The mean annual ensemble evaporation of **(c)** CMRSET, SSEBop and MOD16 ($E_{\mathrm{CSM}}$) for the years 2003–2012 (50° N–50° S), and **(e)** SSEBop and MOD16 ($E_{\mathrm{SM}}$) for the years 2003–2013 (80° N–56° S). Standard deviation of ensemble evaporation of **(e)** $E_{\mathrm{CSM}}$, and **(f)** $E_{\mathrm{SM}}$. Values below 0.5 % of the maximum are displayed as white.

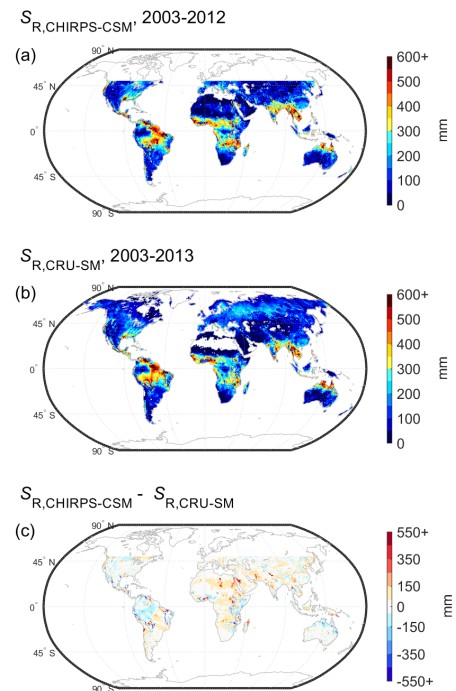

**Figure 3.** Root zone storage capacity estimates of **(a)** $S_{\mathrm{R,CHIRPS-CSM}}$ (based on $P_{\mathrm{CHIRPS}}$ and $E_{\mathrm{CSM}}$), **(b)** $S_{\mathrm{R,CRU-SM}}$ (based on $P_{\mathrm{CRU}}$ and $E_{\mathrm{SM}}$), and **(c)** the difference between $S_{\mathrm{R,CHIRPS-CSM}}$ and $S_{\mathrm{R,CRU-SM}}$. Values below 0.5 % of the maximum in (a) and (b) are displayed as white.

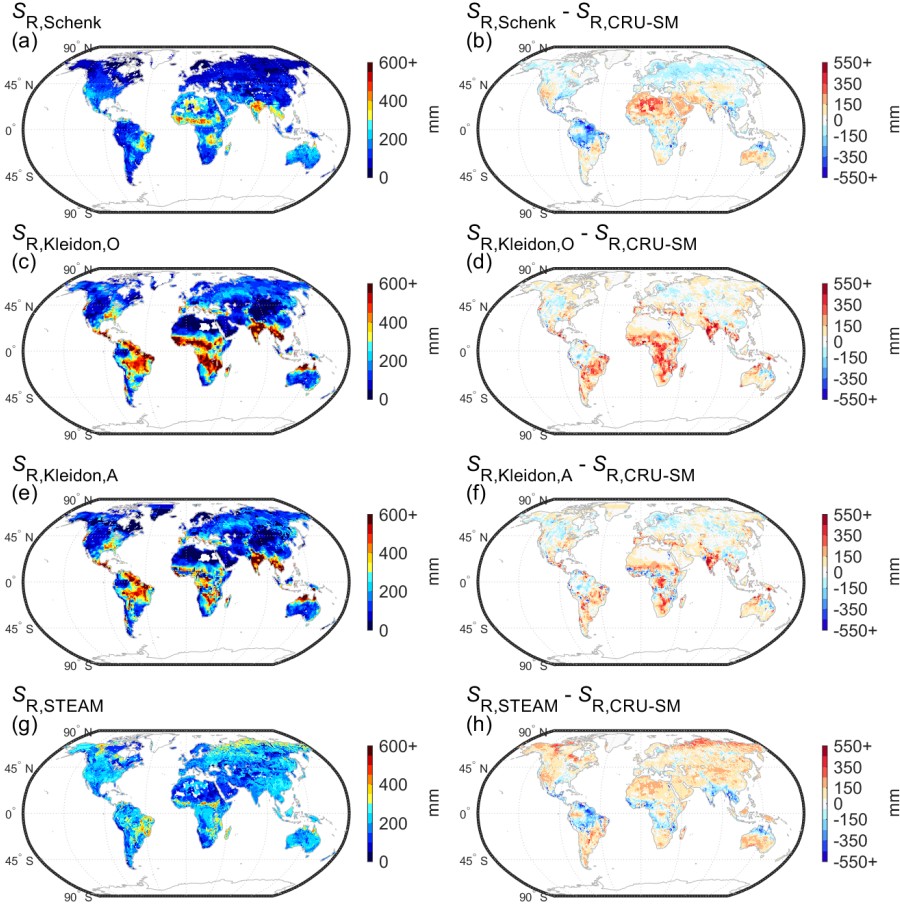

**Figure 4.** Root zone storage capacities of **(a)** $S_{\mathrm{R,Schenk}}$ (Schenk and Jackson, 2009), **(c)** $S_{\mathrm{R,Kleidon,O}}$ (Kleidon, 2004), **(e)** $S_{\mathrm{R,Kleidon,A}}$ (Kleidon, 2004), **(g)** $S_{\mathrm{R,STEAM}}$ (based on look-up table in Wang-Erlandsson et al. (2014)) and **(b, d, f, h)** their differences with $S_{\mathrm{R,CRU-SM}}$ (estimated based on $E_{\mathrm{SM}}$ and $P_{\mathrm{CRU}}$ in this study). Values below 0.5 % of the maximum in (a), (c), (e), and (g) are displayed as white.

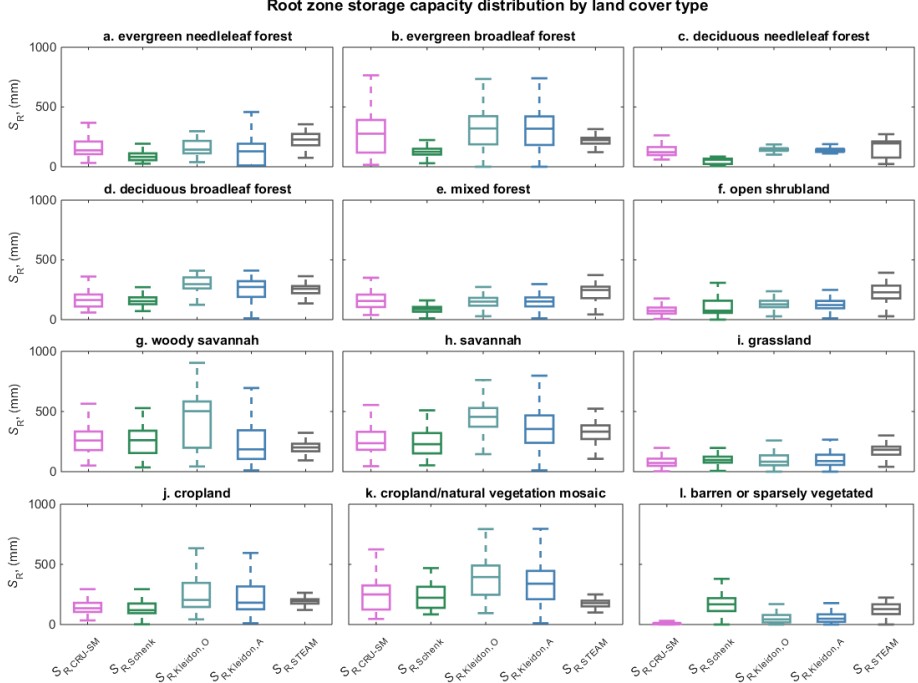

**Figure 5.** Comparison of root zone storage capacity estimates by land cover type using Tukey boxplots. The central markers of the boxes mark the median, and the box edges mark the 25$^{th}$ and 75$^{th}$ percentile. The whiskers extend to 1.5 times the interquartile range.

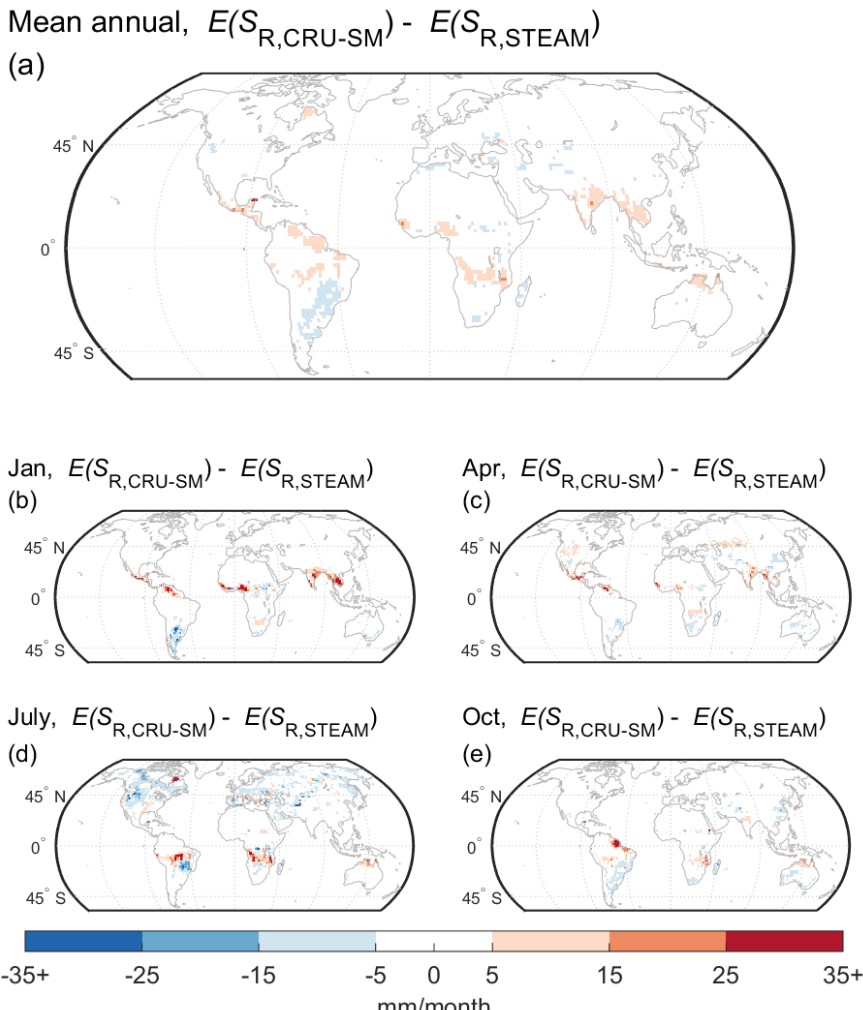

**Figure 6.** Difference in STEAM-simulated evaporation between using $S_{R,CRU-SM}$ (estimated based on $E_{SM}$ and $P_{CRU}$ in this study) and $S_{R,STEAM}$ (based on look-up table in Wang-Erlandsson et al. (2014)) as root zone storage capacity parametrisation at **(a)** mean annual scale and averages for the months of **(b)** January, **(c)** April, **(d)** July, and **(e)** October over the time period 2003–2013. See also Sect. 4.3.

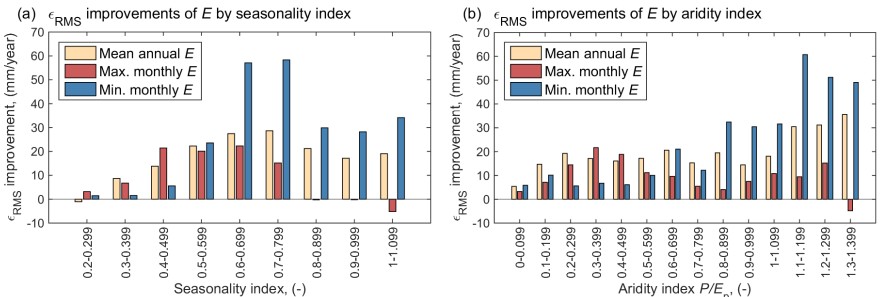

**Figure 7.** The improvement in root mean square error ($\varepsilon_{\text{RMS}}$) in simulated mean monthly evaporation $E$ by implementing $S_{\text{R,CRU−SM}}$ (estimated based on $E_{\text{SM}}$ and $P_{\text{CRU}}$ in this study) instead $S_{\text{R,STEAM}}$ (based on look-up table in Wang-Erlandsson et al. (2014)) in the global hydrological model STEAM. The improvements in mean annual, mean maximum monthly and mean minimum monthly $E$ (over the years 2003–2013) are sorted by **(a)** precipitation seasonality index and **(b)** aridity index (defined in Appendix B1). The satellite based ensemble evaporation based on SSEBop and MOD16 ($E_{\text{SM}}$) was used as the benchmark for improvements, (see methods described in Sect. 2.2). Only bins containing a minimum of 200 grid cells are shown.

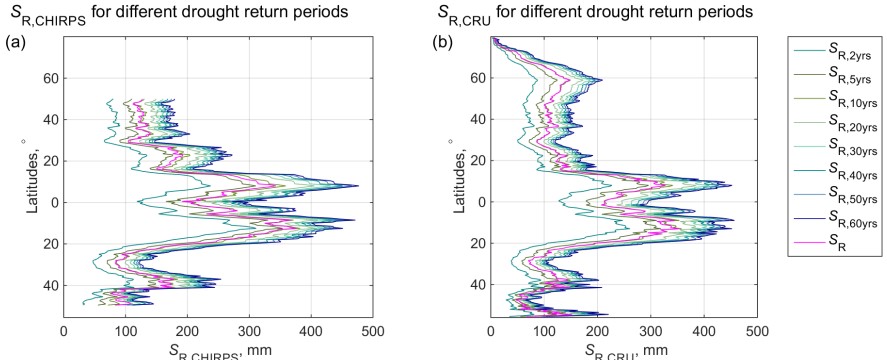

**Figure 8.** Mean latitudinal root zone storage capacity **(a)** $S_{\text{R,CHIRPS−CSM}}$ (based on $P_{\text{CHIRPS}}$ and $E_{\text{CSM}}$) and **(b)** $S_{\text{R,CRU−SM}}$ (based on $P_{\text{CRU}}$ and $E_{\text{SM}}$) dimensioned by drought return periods between 2 and 60 years estimated using Gumbel distribution (see methods described in Sect. 2.3).

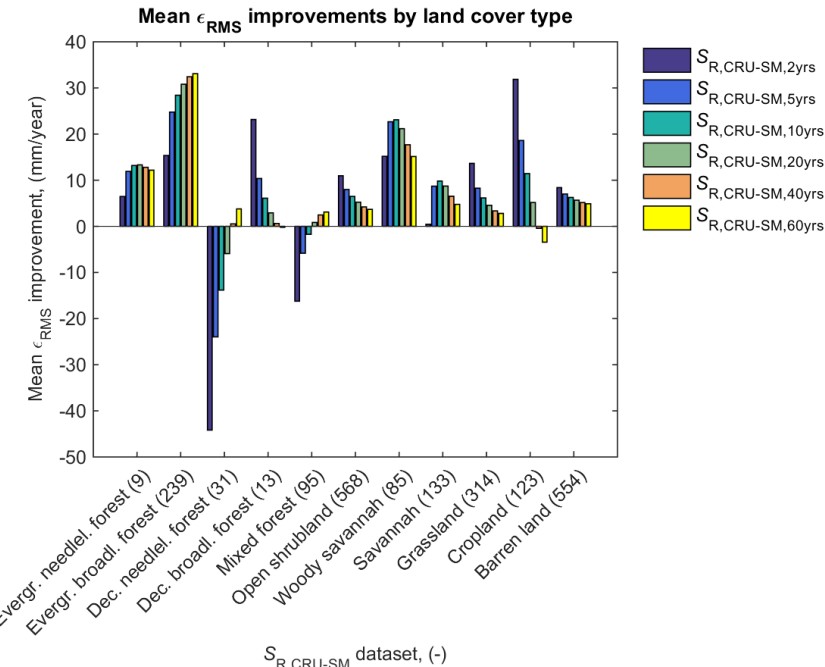

**Figure 9.** The mean $\varepsilon_{\mathrm{RMS}}$ improvement in simulated monthly evaporation $E$ (2003–2013) by implementing $S_{\mathrm{R,CRU-SM},L\mathrm{yrs}}$ (based on $P_{\mathrm{CRU}}$ and $E_{\mathrm{SM}}$) instead of $S_{\mathrm{R,STEAM}}$ (based on look-up table in Wang-Erlandsson et al. (2014)) in the global hydrological model STEAM, where the satellite based $E_{\mathrm{SM}}$ was used as the benchmark for improvements (see methods described in Sect. 2.2). The improvements for root zone storage capacities with different return periods $L$ 2-60 yrs (i.e., $S_{\mathrm{R,CRU-SM,2yrs}}$, $S_{\mathrm{R,CRU-SM,5yrs}}$, $S_{\mathrm{R,CRU-SM,10yrs}}$, $S_{\mathrm{R,CRU-SM,20yrs}}$, $S_{\mathrm{R,CRU-SM,40yrs}}$, and $S_{\mathrm{R,CRU-SM,60yrs}}$) are shown for the different land cover types that has $>90\,\%$ grid cell coverage. The number of represented grid cells are provided in the parenthesis following each land cover type label along the x-axis.

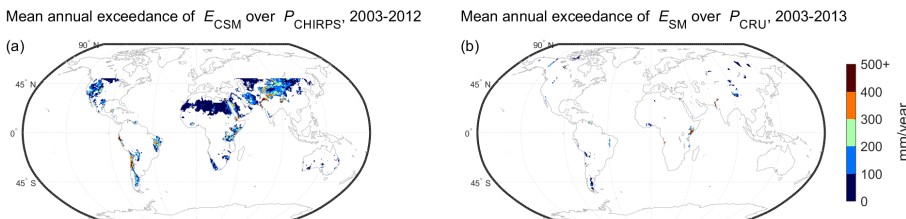

**Figure A1.** Mean annual accumulated exceedance of **(a)** $E_{\mathrm{CSM}}$ (ensemble evaporation of CMRSET, SSEBop, and MOD16) over $P_{\mathrm{CHIRPS}}$, and **(b)** $E_{\mathrm{SM}}$ (ensemble evaporation of SSEBop, and MOD16) over $P_{\mathrm{CRU}}$.

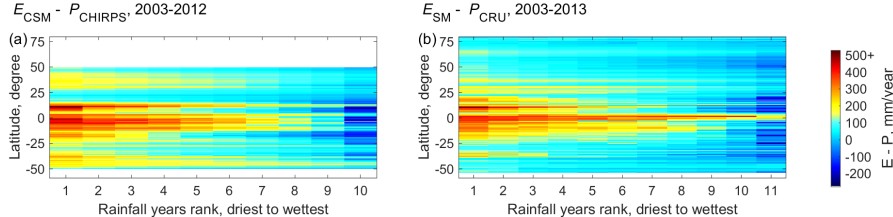

**Figure A2.** Mean latitudinal difference between **(a)** $E_{\text{CSM}}$ (ensemble evaporation of CMRSET, SSEBop, and MOD16) and $P_{\text{CHIRPS}}$ (CHIRPS precipitation), and **(b)** $E_{\text{SM}}$ (ensemble evaporation of SSEBop, and MOD16) and $P_{\text{CRU}}$ (CRU precipitation) sorted from the driest to the wettest years. The figure only includes regions where accumulated $E - P$ over the entire available time series (2003–2012 and 2003–2013 respectively) are positive.

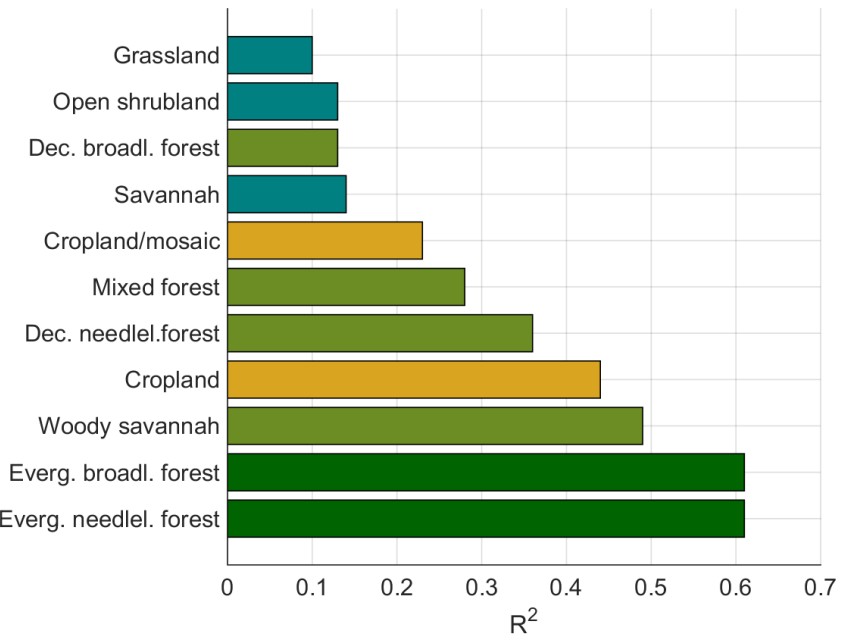

**Figure A1.** Coefficient of determination $R^2$ of the multiple linear regression model of $S_{\text{R,CRU−SM}}$ (based on $P_{\text{CRU}}$ and $E_{\text{SM}}$) based on the climate variables interstorm duration $I_{\text{isd}}$, precipitation seasonality $I_{\text{s}}$, and aridity $I_{\text{a}}$. The green bars are forests or wooded land, the yellow bars represent croplands, and the teal bars represent short vegetation types.