# Peer review of "Global root zone storage capacity from satellite-based evaporation"

_Hydrology and Earth System Sciences, 2015_

## Referee Comment (RC1) · Anonymous Referee #1 · 25 Jan 2016

**OVERVIEW**

The manuscript investigates a method for estimating root-zone storage capacity, $S_R$, by using remote sensing observations. Specifically, satellite-derived evapotranspiration data and gauge/satellite precipitation data are used for estimating $S_R$ on a global scale. The obtained maps of $S_R$ are compared with three previous $S_R$ products and the differences between datasets are analysed. Finally, the new $S_R$ dataset is used in the global hydrological model STEAM for analysing the impact on evapotranspiration

estimation of the new $S_R$ parameterization.

**GENERAL COMMENTS**

The manuscript is well written and quite clear. The topic is really of interest as the estimation of root-zone storage capacity, $S_R$, on a global scale would be highly beneficial for modelling and understanding the land-atmosphere interaction with several implications on climate, agriculture, hydrology, etc. The study build on previous studies from the same authors that were made on regional and/or local scale and here analyses the possibility to estimate $S_R$ on a global scale by using satellite data. I believe the paper deserves to be published for the high relevance of the investigated topic. However, in my opinion, several aspects should be improved/changed before the publication. I reported below a list of the general comments to be addressed with also the specification of their relevance.

1) MAJOR: The description of the method should be improved. Is the method the same as in previous studies (e.g., Gao et al. 2014 GRL)? If yes, it should be clearly acknowledged. Is it different from the paper (under review, not available to reviewers) by Boer-Euser et al.? It should be clear to the reader if the novelty of the papers is on the method or in the satellite dataset used as input. Please clarify.

2) MODERATE: In the "Methods" section it reads several assumptions: (i) "irrigation is captured in satellite-based evaporation data", I am not sure it is true. At least, not for all satellite-based datasets, please clarify. Moreover, at page 14 it reads that the evaporation originating from irrigation water simulated by LPJmL is considered. Why if irrigation is already included in the satellite-based evaporation data? (ii) "the long term average is added ... in order to compensate for overestimation of evaporation and underestimation of precipitation". Why? (iii) "in order to take into account of surface

Interactive
comment

runoff, D never becomes negative". Again, why?

I believe that the authors should better justify the assumption made in their method and why these assumptions are valid (or not). This will allow the reader to understand the strengths and the limitation of the proposed approach.

3) MAJOR: The selection of the input datasets is for me a major issue. Again, it should be clarified why satellite-based data are considered for evaporation and not for precipitation. Why satellite-based datasets for precipitation are not considered (e.g., TMPA, CMORPH, PERSIANN)?

Moreover, why the average of the three evaporation datasets should be "attractive"? Are the results changing by using only one of the datasets? What is the relative impact of the evaporation and precipitation datasets on the final results? It should be clarified, too.

Why ERA-Interim data are used for temporal downscaling? Apart that it is not mentioned how the temporal downscaling is carried out, currently daily evaporation and precipitation datasets are (freely) available (actually, several datasets). Why the authors do selected monthly datasets and then performed downscaling with ERA-Interim? Why not using directly ERA-Interim data? Or other daily products (e.g., GLEAM for evaporation and TMPA for precipitation)? All these points should be clarified.

4) MODERATE: In most of the paper, only the $S_{R_{CRU-SM}}$ dataset is analysed. Why two datasets are considered (CHIRPS and CRU)? The real value of considering also the CHIRPS dataset is not clear to me. Please clarify.

5) MAJOR: I found the selection of the application for validating the obtained $S_R$ maps not correct. In the paper, it is assessed the improvement in estimating evaporation with the new $S_R$ parameterization in STEAM. It is fine for me. The problem is that the same evaporation dataset ($E_{SM}$) used for computing $S_R$ is also used for assessing the improvements due to the new $S_R$ parameterization. It is a circular argument that

is not good. I suggest performing a different validation test. Why not considering the differences in the runoff prediction with the old and new $S_R$ parameterization? It looks to me much more relevant, and a good independent evaluation.

6) MINOR: A very recent paper with the same topic has been published in Journal of Hydrology by Campos et al. (2016, http://dx.doi.org/10.1016/j.jhydrol.2016.01.023). I suggest mentioning and analysing this study.

In the specific comments, I added some corrections and suggestions that should be implemented.

On this basis, I believe the paper deserves to be published only after a major revision.

**SPECIFIC COMMENTS (P: page, L: line or lines)**

P10, L13: "The resetting of this limited number of pixels..". Please specify what is the percentage number of pixels for which resetting was needed.

P10, L19-23: Simply say that $S_R$ is the maximum of the obtained D values.

P11, L8-11: This paragraph is not clear to me, please revise.

P12, L10: The C parameter values is set to a value equal to 0.1. Why? What is the impact on the results? Why "C" in equation (7) is different from "c" in equation (6)?

P12, L18: $S_{R,CRU-SM}$ is not defined, only later in the text.

P12, L20: The formulation of equation (8) is wrong for me. The root mean square error should be between $E_{S_{R,STEAM}}$ and $E_{SM}$, not only for $E_{S_{R,STEAM}}$ or $E_{SM}$. Please reformulate.

P13, L16-17 The symbols $sigma_n$ and $sigma_{S_R}$ are missing in the text. Please correct.

[Figure]

P16, L13: ERA-I evaporation is used as forcing of STEAM. What is the output of STEAM? Is it the actual evaporation? It should be clarified and clearly distinguished from potential evapotranspiration throughout the text.

P16, L27-28: The methods used for downscaling/upscaling the different datasets should be described.

P17, L12: "$S_R$ estimated are generally larger", larger than? Please clarify.

P19, L22-P20, L8: Too many details are given here for the description of the differences of the simulated evaporation data. It is difficult to follow, please reduce the text focusing on the most relevant differences.

P22, L20: Recent studies have obtained huge differences between global scale precipitation datasets (e.g., *Trenberth et al. (2014)*, *Herold et al. (2016)*). It seems not true that evaporation data (on a global scale) have larger spread than precipitation data. Please reformulate.

P25, L13: It seems that "and $S_{R_{CRU-SM}}$" is missing here.

Tables/Figures: Please check captions for symbols. Captions should be self-describing.

**REFERENCES**

Herold, N., et al. "How much does it rain over land?" Geophysical Research Letters in press (2016). DOI: 10.1002/2015GL066615.

Trenberth, K.E., et al. "Global warming and changes in drought." Nature Climate Change 4.1 (2014): 17-22.

---

## Referee Comment (RC2) · Anonymous Referee #2 · 27 Jan 2016

The manuscript "Global root zone storage capacity from satellite-based evaporation" from Wang-Erlandsson et al, submitted to Hydrology and Earth System Sciences, describes an interesting approach of filling a gap in global-scale water modelling by developing a method as well as a dataset for global-scale root zone storage capacity. The study is based on the assumption that plants are optimizing their roots to drought conditions (with a given return period). Based on the cumulative difference of the sum of (satellite and ground observation based) precipitation and modelled irrigation and two to three MODIS based evaporation products, the authors used the largest "drought" (soil moisture deficit) condition to assess the root zone storage capacity. Furthermore, they used the Gumbel-approach to assess vegetation types with longer return periods than the data provides as well for a sensitivity analysis. The resulting product is compared to different types of previous work as well as used in a model experiment.
[Figure]

First of all, the paper is well written, well structured and I really like the review of methods to estimate the root zone capacity. I think the overall approach can contribute in better quantifying worlds water resources with global model approaches. Anyhow, there are some (mostly minor) drawbacks that should be improved before publishing.

General comments

The beginning of Sect. 2.1 should be revised. It refers to a figure which is (for me) not really self-explaining. Also, I had problems while reading Page 9, line 10 "P and the evaporation originating form irrigation". I asked myself why is this evaporation not included in the F_out term. I suggest to write something like "the amount of effective irrigation water (that is evapotranspired by the crops)".

At Section 4.3 the authors assess the improvement of the new root zone storage capacity information within the STEAM model by comparing it to the same product with which the root zone storage capacity was developed. I wonder if this is an independent benchmark product or if you should use E_CSM instead (or only E_C) as benchmark. Sure, there is a lack of real observation based benchmark products for evaporation, but this is a weak point. You should select a different benchmark or rephrase the term benchmark.

Most of the figures are very tiny, and sometimes due to the choice of color very hard to read (e.g. Fig. A1). Please take care of figure size in the final production phase of the manuscript if it is accepted.

Specific comments

I suggest changing the first word of the title to Global-scale. That is more reflecting the issue, that the product is done for global land surface (but one can also live with just "global")

Page 6, line 23: I would recommend a new section starting with "For global" or revising the first sentence. The last sentence of the section are general statements of model

calibration, but to my knowledge, none of the models described here is calibrated directly on root zone storage capacity as it is written in the first sentence. Please revise.

Page 8, line 2: you should change "stores" to "storages" as this is more common term in that sense

Page 8, line 10: change "global" by "global-scale"

Page 9, line 15 f: strange sentence. Why should one understand that irrigation is included in precipitation data? Or do you refer to satellite precipitation data? This sentence needs to be revised

Page 10, line 13: please be precise and present the number of pixels that are affected.

Page 11, line 21: a dot at end of section is missing.

Page 13,line 3: I wonder how many of the 1.5 deg grid cells are available for each land cover class, if land cover needs to be at least 90% of a single land cover. Maybe I have misinterpreted the information, but is it correct that you used MODIS land cover with 0.05° resolution to assess land cover for the 1.5° cell? So, the 1.5°cell consists of 90 0.05° tiles, and at least 81 of them needs to be in one land cover class. The global pattern of land cover is very heterogeneous and think it is important for interpreting the results if you write (e.g. in a table) the number of grid cells per land cover class that went into that analysis.

Page 15, line 8 ff: please describe at least in some sentences the temporal downscaling of monthly evaporation and precipitation data.

Page 16, line 24: again, it would be nice to have an idea, how many grid cells are used per land cover class.

Page 20, line 14 f: Hard to judge that because de Boer-Euser is not available for the reviewers. Maybe you should write in a few sentences what is written there.

Page 21, line 14 f: The term "speculate" is not nice in a paper. Is there any reference

for the message "deciduous forests need a large root zone storage capacity to cater for dry periods during their most active summer months" available or is it something you found out during your research? Please try to verify this. Same for the next sentence.

Page 23, line 2: after "from" can be misinterpreted and is not complete. I suggest to write: "...from remotely sensed evaporation, remotely sensed and station based precipitation and model based irrigation..."

Page 23, line 7: did you compare SR,chirps-CSM with E_SM or with E_CSM? Is that conclusion somewhere covered in the paper?

References, general: why are the page numbers written after the reference? Is this the new standard of HESS?

Page 28, line 12: belongs "Open Access" really to the journal title?

Page 31, line 3: Please check the citation. It is a master thesis, and I am not sure if there are so many co-authors.

Page 31, line 22: Check names

Page 32, line 4: does Jennings really have CMHCMH as initials? I tried to get access but that failed. Could you maybe update the resource?

Page 33, line 24: soil should be in lower case

Page 35, line 8: check initials from last co-author

Page 40, Fig 3: For Africans desert region, the CRU-SM product has obviously values of 0, whereas the CHIRPS-CSM product is > 0, and if I see it correctly at Fig 3c, it has reasonable large values. Could you explain somewhere, where the difference comes from - is it due to precip product or due to additional evaporation product?

Page 44, Fig 7: please write after aridity index that the calculation can be found in Sect B1.

Page 45, Fig. 8: Due to the legend, both subfigures does not have the same width. Please try this for better comparability.

---

## Referee Comment (RC3) · A. Kleidon (Referee) · 12 Feb 2016

This manuscript describes the application of a novel approach to infer root-zone water storage capacity from an evaporation data set derived from remote sensing and illustrates the consequences in a hydrologic model. The manuscript is generally very well written, novel and innovative, and should be accepted for publication after addressing a few minor comments.

I only have a few minor comments:

- page 4-6: The introduction contains a nice review of the different methodologies. I think for what you describe as the "root distribution modelling approach" is more appropriately labeled an "optimization/maximization approach", as it infers rooting properties from some ecological cost function. Also, Kleidon and Heimann 1998 did not use an

inverse approach, but an optimization approach, even though in a highly simplified way and without the use of an explicit root distribution, so it may be better to refer to such a class of "optimization" approaches.

- page 7, line 10: I think more relevant is here a link to cost-benefit type of analysis rather than evolution. It may be appropriate to refer to the classic book edited by Givnish "On the economy of plant form and functioning" by Cambridge University Press.

- page 10, line 14: "in any measureable way" sounds rather strong. Perhaps better to say that it only affects results to a small extent?

- page 12, line 12: It would be nice to see how well the two formulations of the stress function compare to each other. Can you show this in a figure?

- page 14, line 14: "electro-magnetic spectrum" — do you mean different wave-lengths/bands?

- page 16, line 14: "wind speed in two directions" — really? If so, why do you use both directions? Or do you use the two measurements to calculate wind speed?

- page 18, lines 4-11/Figure 4: the correspondence (or disagreements) between the data sets would be easier to see in a scatterplot, where the different data sets are compared at a grid-by-grid scale. How well they correspond is then reflected by the slope of the regression as well as the r2 value. It is probably not necessary to show all scatterplots (or add them as supplementary), but I think this type of analysis would really help to identify how well the different data sets compare to each other.

- page 18, lines 12-28: In the discussion of the differences, it is also important to note that these datasets may use different climate data sets, particularly precipitation. Also, Kleidon (2004) calculated evaporation in a quite simple way, which also is likely to result in differences. What this means is that the differences may not simply reflect on different ways to infer rooting properties, but there is also a component related to the forcing datasets which is difficult to quantify.

- page 25, line 2: Note that for the effect of climate change, it also depends on the ability of vegetation to adapt to altered conditions. This aspect should be mentioned.

- page 38, Figure 1: This figure nicely illustrates the concept. I think it could be made even better if you show the integrated fluxes of Fin-Fout over time in a separate plot above the panel where you show the bins.

- page 40, Figure 3: Start the caption more descriptive with something like "Estimates of root zone storage capacity of the ....". You may also want to use the same color scale in panel (c) as in Fig. 4 to facilitate comparison?

- page 43, Figure 6: I find the differences difficult to see. It may be easier to attribute the differences when you use only a few discrete color values with less than 8 shadings so that one more clearly associate the differences in a region with the values. (also applies to other plots)

———————————————

---

## Author Comment (AC1) · 22 Feb 2016

We are grateful for the constructive comments, and would here like to briefly respond to Referee #1's general comments. The specific comments will be addressed in a later response. Below, referee #1's general comments are in bold and our responses are in upright font. We refer to the manuscript for explanations of variables and abbreviations.

**1) MAJOR: The description of the method should be improved. Is the method the same as in previous studies (e.g., Gao et al. 2014 GRL)? If yes, it should be clearly acknowledged. Is it different from the paper (under review, not available to reviewers) by Boer-Euser et al.? It should be clear to the reader if the novelty of the papers is on the method or in the satellite dataset used as input. Please clarify.**

[Figure]

We consider this paper to be novel in several ways, not isolated to either method or satellite dataset input. The novelties include:

- showing that a mass balance based method for estimating $S_R$ is suitable for global scale application,

- making use of the recent developments in satellite-based evaporation data to estimate $S_R$ at the global scale,

- present a $S_R$ dataset that can be used directly by the community of global modellers,

- providing significant new insights on the strategy of natural vegetation to deal with droughts and being resistant to future climate fluctuations,

- showing how evaporation simulation changes when conventional look-up table derived root zone storage capacity information is replaced by the presented $S_R$ dataset, and showing how this compares to independent evaporation products, and

- investigating the differentiated drought return periods for different land use types.

While we make use of the same mass balance principle as applied by Gao et al., (2014) and de Boer-Euser et al., (2016) (now published), our algorithm is based on indirect measurements of every unique pixel that reduces various assumptions related to atmospheric exchanges between land and atmosphere. The scale of application, and input data are different as well.

Importantly, Gao et al., (2014) did not use actual evaporation data as input, and used NDVI to estimate the slope of transpiration during different seasons. The slope of transpiration was kept constant, and the mass balance was calculated using the Mass

Curve Technique, and not a running "deficit" account as in this paper. In contrast to the method used by de Boer-Euser et al. (2016), we do not a priori assume a Gumbel return period. Moreover, both Gao et al., (2014) and de Boer-Euser et al. (2016) applied the mass balance approach at the catchment scale and estimated interception in order to use transpiration with effective precipitation. In this paper, we show that it is feasible to apply the concept directly at the global scale using independent pixel measurements instead of catchments as units and simplified approximations of total evaporation and total precipitation. An overview of the differences is provided in Table 1.

Although the basics of both methods (Gao et al., 2014; de Boer-Euser et al., 2016) were covered in the Introduction, we agree that the differences could be made clearer. In the revision, we will highlight the novelties of this paper, especially attending to the differences between our method and those applied in Gao et al., (2014) and de Boer-Euser et al., (2016).

**2) MODERATE: In the "Methods" section it reads several assumptions: (i) "irrigation is captured in satellite-based evaporation data", I am not sure it is true. At least, not for all satellite-based datasets, please clarify. Moreover, at page 14 it reads that the evaporation originating from irrigation water simulated by LPJmL is considered. Why irrigation is already included in the satellite-based evaporation data? (ii) "the long term average is added in order to compensate for overestimation of evaporation and underestimation of precipitation". Why? (iii) "in order to take into account of surface runoff, $D$ never becomes negative". Again, why? I believe that the authors should better justify the assumption made in their method and why these assumptions are valid (or not). This will allow the reader to understand the strengths and the limitation of the proposed approach.**

(i) Satellite-based evaporation datasets determine the latent heat flux from radiation data and thermal infrared data. The origin of evaporation cannot be determined, although it can be verified whether $E$ is exceeding net precipitation $P_{\text{net}}$. Several papers have demonstrated the capability to distinguish total $E$ into the component that originates from rainfall $P$ and from other sources such as irrigation, inundations and groundwater dependent ecosystems (e.g. Ahmad et al., 2005; Van Eekelen et al., 2014; Bastiaanssen et al., 2014). The latent heat flux simply encompasses all these processes, hence, evaporation from irrigation is implicitly incorporated. Because of this, we must remove it from our $S_R$ calculations, and therefore used irrigation evaporation from LPJmL. Other models such as GlobWat (Hoogeveen et al., 2015) and Water Footprint (Chukalla et al., 2015) could have been used alternatively. If evaporation irrigation is not removed, the $S_R$ estimates in irrigated regions would be greatly overestimated, as we would have mistaken irrigation water to come from a natural soil moisture store. We wrote at P. 9 L. 16-18: "Without correction, the irrigation evaporation in the satellite evaporation data would erroneously contribute to accumulation of soil moisture deficit in our computations."

(ii) "The long-term average is added..." in certain grid cells where accumulated $E$-$P$ is positive for several years in a row, in order to prevent $S_R$ from growing every year. Continuously increasing $E$-$P$ may be linked to wetlands, irrigated fields or natural seepage zones, where lateral inflow of water occurs. In our method this would lead to accumulation of moisture, which is not possible over longer periods of time. Some deeply rooting vegetation may accumulate moisture over more than a year to replenish moisture deficits of previous years, but positive $E$-$P$ values over several years need to be compensated. For clarification, we will revise at P. 9 L. 20-21: "[the long term average]... [is added] to the inflow, in order to compensate for lateral inflow or estimation errors in evaporation or precipitation."

(iii) We assume that any excess precipitation that cannot be contained by the root zone storage reservoir (or $D$, which is essentially a running estimate of root zone storage reservoir size) is runoff or recharge. Since the reservoir size is calculated through $D$, $D$ can never be negative. We wrote that this is a way to take into account surface runoff, but it could also be expressed as the inevitable procedure if $D$ is defined as the running

estimate of the root zone storage reservoir size.

We will make these points clearer in the revised manuscript.

**3) MAJOR: The selection of the input datasets is for me a major issue. Again, it should be clarified why satellite-based data are considered for evaporation and not for precipitation. Why satellite-based datasets for precipitation are not considered (e.g., TMPA, CMORPH, PERSIANN)?**

**Moreover, why the average of the three evaporation datasets should be "attractive"? Are the results changing by using only one of the datasets? What is the relative impact of the evaporation and precipitation datasets on the final results? It should be clarified, too.**

**Why ERA-Interim data are used for temporal downscaling? Apart that it is not mentioned how the temporal downscaling is carried out, currently daily evaporation and precipitation datasets are (freely) available (actually, several datasets). Why the authors do selected monthly datasets and then performed downscaling with ERA-Interim? Why not using directly ERA-Interim data? Or other daily products (e.g., GLEAM for evaporation and TMPA for precipitation)? All these points should be clarified.**

To estimate $S_{\mathrm{R}}$, we need global coverage at a grid cell resolution for both evaporation and precipitation. Importantly, these products that must not be produced using assumptions on root zone storage capacity, to prevent circularity (since we are estimating root zone storage capacity). In other words, there should be no water balance type of computation process involved in the determination of $S_{\mathrm{R}}$. We used satellite-based evaporation products because they are the only options available that fulfil these criteria, i.e., reanalyses and land surface model evaporation contain soil depth information. Flux net data are too sparse for acquiring consistently good quality global coverage). Conversely, precipitation data do not need to be satellite-based, but can also be ground-based. In this manuscript, we used CRU (ground-based) and CHIRPS

(satellite-based).

Inter-comparison of precipitation products show that both CRU and CHIRPS are good quality precipitation products. In particular, CHIRPS performance stands out in a comprehensive inter-comparison of 13 difference precipitation products in the Nile basin (Hessels, 2015). PERSIANN was the worst performer in this inter-comparison, whereas CMORPH performed the worst in several bias analyses (Hessels, 2015). On-going analyses (not yet published) in the Mekong basin, Vietnam and Colombia also show that CHIRPS performs excellently, so the CHIRPS performance in three continents are outweighing the older technologies used for PERSIANN, CMORPH and APRODITE. One paper on Vietnam is currently under review elsewhere (Simons et al., in review).

The average of the three evaporation datasets is the simplest and most transparent way to create an ensemble product. (Hofste, 2014) also showed that three different approaches towards an ensemble evaporation product (i.e., simple averaging, expert judgement, and outlier removal based on MODIS16NBI, SSEBop, CMRSET, and ALEXI7) all performed similarly better than the individual constituent evaporation datasets in the Nile basin. In their comparison analyses, the mean product outperformed the one based on expert judgement and exhibited less "worst sub-basin performances" than their outlier analysis product, resulting them to conclude that the mean ensemble without any imposed restrictions is "relatively safe" to use.

We have performed analyses of the relative impact of evaporation and precipitation, which we did not include in the paper for conciseness. However, the referee reminds us that this could be of interest for the reader, and we will therefore include these results in the Supplements in the revised version.

Daily remotely-sensed ALEXI-based $E$ data is under production by NOAA and USDA. Daily data based on VIIRS will be made available. The new ETMonitor remote sensing algorithm from the Chinese Academy of Sciences also envisages daily fluxes at

the global scale, but at this moment these products are not available yet. There are currently (February 2016) seven global remote sensing products of $E$ available. The monthly evaporation data used in the manuscript were those available at the time of this research. We did not consider using GLEAM, since the evaporation data is modelled with a-priori assumptions on soil layer depth by land use type (Miralles et al., 2011). For the same reasons, we did not employ the ERA-Interim evaporation, since it is based on the land surface model TESSEL (van den Hurk et al., 2000).

In the temporal downscaling, we first established the ratios between daily values to the mean monthly ERA-Interim, and second, used the relationship to estimate daily values from monthly $E_{\mathrm{SM}}$ or $E_{\mathrm{CSM}}$ values. We will describe this procedure in the revised manuscript.

**4) MODERATE: In most of the paper, only the $S_{\mathrm{R,CRU\text{-}SM}}$ dataset is analysed. Why two datasets are considered (CHIRPS and CRU)? The real value of considering also the CHIRPS dataset is not clear to me. Please clarify.**

We present CHIRPS combined with the mean of SSEBop, CMRSET and MODIS, because CHIRPS is the lead precipitation product and has a fine resolution of 5 km pixels. The use of three evaporation datasets decrease uncertainties related to individual evaporation products because there is simply not one single preferred model. Research executed by Hofste (2014) for the Nile basin demonstrated that the performance of an ensemble $E$ product is significantly better than using individual $E$ products, something that was confirmed by Simons et al. (in review) in Vietnam. However, CHIRPS is unfortunately not available at the global scale, and CMRSET is not reliable in high latitudes, and the modelling community likely needs a global $S_{\mathrm{R}}$ product. Thus, we added $S_{\mathrm{R,CRU\text{-}SM}}$. This way, we have a global $S_{\mathrm{R}}$ map that can be compared to the $S_{\mathrm{R,CHIRPS\text{-}CSM}}$ for reference. We will clarify this further in the revised manuscript.

Note also that we show results of the CHIRPS analyses in the Supplementary Information. We explained (P. 19 L. 17-18) that we did not present $E$ simulation results

using $S_{\mathrm{R,CHIRPS\text{-}CSM}}$ in the main manuscript, since it does not have global coverage, but referred to the Supplementary Information for a comparison between using $S_{\mathrm{R,CRU\text{-}SM}}$ and $S_{\mathrm{R,CHIRPS\text{-}CSM}}$.

**5) MAJOR: I found the selection of the application for validating the obtained $S_{\mathbf{R}}$ maps not correct. In the paper, it is assessed the improvement in estimating evaporation with the new $S_{\mathbf{R}}$ parameterization in STEAM. It is fine for me. The problem is that the same evaporation dataset ($E_{\mathbf{SM}}$) used for computing $S_{\mathbf{R}}$ is also used for assessing the improvements due to the new $S_{\mathbf{R}}$ parameterization. It is a circular argument that is not good. I suggest performing a different validation test. Why not considering the differences in the runoff prediction with the old and new $S_{\mathbf{R}}$ parameterization? It looks to me much more relevant, and a good independent evaluation.**

We consider validation using $E_{\mathrm{SM}}$ to be appropriate, since the algorithms for estimating $S_{\mathrm{R}}$, and for estimating $E$ in STEAM are very different. First, $S_{\mathrm{R}}$ is derived based on the $E$ overshoot over $P$, whereas STEAM is a process-based model where evaporation originates from five different compartments, each constrained by potential evaporation and related stress functions. This means that it is impossible to reproduce $E_{\mathrm{SM}}$ simply by inserting $S_{\mathrm{R}}$ to STEAM. If $S_{\mathrm{R}}$ is zero because $E_{\mathrm{SM}}$ never overshoots precipitation, STEAM soil evaporation and transpiration would become zero. If extreme $S_{\mathrm{R}}$ are produced because $E_{\mathrm{SM}}$ is unrealistically large, STEAM evaporation will not approach $E_{\mathrm{SM}}$, since it will be capped by potential evaporation. Second, consider also that the precipitation products (CRU and CHIRPS respectively) used for deriving $S_{\mathrm{R}}$ differ from the precipitation forcing (ERA-Interim) used in STEAM. Third, $E_{\mathrm{SM}}$ and STEAM are truly independent to each other as well. Whereas STEAM is process and water balance based, the ensemble $E$ product is based on a combination of two($E_{\mathrm{SM}}$)/three($E_{\mathrm{CSM}}$) well established energy balance methods. The only difference of the new STEAM simulations is the inclusion of updated information on root zone storage so that during longer periods of drought, more realistic estimations of continued evaporation processes can be expected. Last, $S_{\text{R,CRU-SM}}$ is based on a single year value of $E_{\text{SM}}$ (i.e., the year of maximum storage deficit), whereas the analyses of improvements were based on the entire available time series of 10-11 years. Thus, the fact that $S_{\text{R,CRU-SM}}$ dimensioned on one year of $E_{\text{SM}}$ nevertheless improves $E$ simulation in STEAM with regard to 10-11 years of $E_{\text{SM}}$ (i.e., the overall $\epsilon_{\text{RMS}}$ decreases when $S_{\text{R,CRU-SM}}$ is used in STEAM) is a strong indication that the storage capacity correction was implemented for the right reason. We maintain that the comparison with $E_{\text{SM}}$ is useful and will clarify our arguments in the revised manuscript. Note also that STEAM is not calibrated by any means.

To address the referee's concern of interdependency, we cross-check the mean monthly STEAM evaporation based on $S_{\text{R,CRU-SM}}$ (2003-2013) with the mean monthly LandFlux-EVAL diagnostic ensemble evaporation (1989-2005) (Mueller et al., 2013), see comparison in Table 2. It appears that the $\epsilon_{\text{RMS}}$ improvements are even greater (mean improvement 10 mm/year instead of 4 mm/year), but with the greatest improvements in maximum monthly evaporation instead of minimum monthly evaporation. The LandFlux-EVAL diagnostic product include the evaporation products: PRUNI, MPIBGC, CSIRO, GLEAM, and AWB. Since this product includes GLEAM, which relies on water balance calculations and soil layer depth assumptions, we consider the use of this product inappropriate for our purposes and would refrain from including this comparison in the manuscript.

Runoff data represent a catchment or basin average value that is not sufficient for validating a spatial map on $S_{\text{R}}$. Using runoff for validation, we would for example not be able to analyse the $E$ simulation performance with regard to climate indicators or land use types. Runoff results are also highly sensitive to the precipitation data used, especially in wet regions (Fekete et al., 2004; Materia et al., 2010). River flow data can be considerably unreliable in several large river basins such as the Congo (Tshimanga and Hughes, 2014). Nevertheless, we have compared simulation results ($P$-$E$) to annual mean GRDC runoff data, and will add it to the Supplementary Information for

readers' reference.

**6) MINOR: A very recent paper with the same topic has been published in Journal of Hydrology by Campos et al. (2016, http://dx.doi.org/10.1016/j.jhydrol.2016.01.023). I suggest mentioning and analysing this study.**

We thank the referee for this suggestion. We will add this to the revised manuscript.

**References**

de Boer-Euser, T., McMillan, H. K., Hrachowitz, M., Winsemius, H. C. and Savenije, H. H. G.: Influence of soil and climate on root zone storage capacity, Water Resour. Res., n/a–n/a, doi:10.1002/2015WR018115, 2016.

Fekete, B., Vörösmarty, C., Roads, J. and Willmott, C.: Uncertainties in precipitation and their impacts on runoff estimates, J. Clim., 17, 294–304, doi:10.1175/1520-0442(2004)017<0294:UIPATI>2.0.CO;2, 2004.

Gao, H., Hrachowitz, M., Schymanski, S. J., Fenicia, F., Sriwongsitanon, N. and Savenije, H. H. G.: Climate controls how ecosystems size the root zone storage capacity at catchment scale, Geophys. Res. Lett., 41(22), 7916–7923, doi:10.1002/2014GL061668, 2014.

Hessels, T. M.: Comparison and Validation of Several Open Access Remotely Sensed Rainfall Products for the Nile Basin, Delft University of Technology., 2015.

Hofste, R. W.: Comparative analysis of near-operational evapotranspiration products for the Nile basin based on Earth Observations; First steps towards an ensemble ET product, Delft University of Technology. [online] Available from: http://repository.tudelft.nl/assets/uuid:16659a39-3256-4ff9-9930-

81ac4dfb4018/maindoc.pdf, 2014.

van den Hurk, B. J. J. M., Viterbo, P., Beljaars, A. C. M. and Betts, A. K.: Offline validation of the ERA40 surface scheme. [online] Available from: http://www.knmi.nl/publications/fulltexts/tm295.pdf (Accessed 13 July 2012), 2000.

Materia, S., Dirmeyer, P. a., Guo, Z., Alessandri, A. and Navarra, A.: The Sensitivity of Simulated River Discharge to Land Surface Representation and Meteorological Forcings, J. Hydrometeorol., 11(2), 334–351, doi:10.1175/2009JHM1162.1, 2010.

Miralles, D. G., Holmes, T. R. H., De Jeu, R. A. M., Gash, J. H. C., Meesters, A. G. C. A. and Dolman, A. J.: Global land-surface evaporation estimated from satellite-based observations, Hydrol. Earth Syst. Sci., 15(2), 453–469, doi:10.5194/hess-15-453-2011, 2011.

Mueller, B., Hirschi, M., Jimenez, C., Ciais, P., Dirmeyer, P. A., Dolman, A. J., Fisher, J. B., Jung, M., Ludwig, F., Maignan, F., Miralles, D. G., McCabe, M. F., Reichstein, M., Sheffield, J., Wang, K., Wood, E. F., Zhang, Y. and Seneviratne, S. I.: Benchmark products for land evapotranspiration: LandFlux-EVAL multi-data set synthesis, Hydrol. Earth Syst. Sci., 17(10), 3707–3720, doi:10.5194/hess-17-3707-2013, 2013.

Tshimanga, R. M. and Hughes, D. A.: Basin-scale performance of a semidistributed rainfall-runoff model for hydrological predictions and water resources assessment of large rivers: The Congo River, Water Resour. Res., 50(2), 1174–1188, doi:10.1002/2013WR014310, 2014.

Simons, G.W.H., W.G.M. Bastiaanssen, L.A. Ngô, C.R. Hain, M.C. Anderson and G. Senay, 2016. Integrating open-access satellite-derived global data products as a pre-analysis for hydrological modelling studies: a case study for the Red River Basin, MDPI Remote Sensing (in review)

Hofste, R.W., 2014. Comparative Analysis Among Near-Operational Evapotranspiration Products for the Nile Basin Based on Earth Observations, M.Sc. thesis, Faculty of

Civil Engineering and Geosciences, Delft University of Technology: 24 pp

Eekelen, van, M., W.G.M. Bastiaanssen, C. Jarmain, B. Jackson, F. Fereira, J. Bosch, P. Dye, P. Van der Zaag, A. Saraiva, E. Bastidas-Obando, R. Dost and W. Luxemburg, 2015. A novel approach for estimating direct and indirect water withdrawals using satellite measurements: a case study from the Incomati Basin, Agriculture, Ecosystems and Environment, 200: 126-142

Ahmad, M.D., W.G.M. Bastiaanssen and R.A. Feddes 2005. A new technique to estimate net groundwater use across large irrigated areas by combing remote sensing and water balance approaches. Hydrogeology Journal Volume 13. Number 5-6: 653-664.

Chukalla, A.D., M.S. krol and A.Y. Hoekstra, 2015. Green and blue water footprint reduction in irrigated agriculture: effect of irrigation techniques, irrigation strategies and mulching, Hhydr. Earth Syst. Sci. 19: 4877-4891

Hoogeveen, J., J.-M. Faurès, L. Peiser, J. Burke, and N. van de Giesen, 2015. GlobWat – a global water balance model to assess water use in irrigated agriculture, Hydrol. Earth Syst. Sci., 19, 3829-3844

**Table 1.** Overview of the methods used in Gao et al., (2014), de Boer-Euser et al. (2016), and this study.

| | Gao et al., (2014) | de Boer-Euser et al., (2016) | This paper |
|---|---|---|---|
| **Scale/Coverage** | Catchment | Catchment | Global |
| **Unit** | Catchment | Catchment | Grid cell (0.5 degree) |
| **Water demand input in the mass balance calculation** | The slope of cumulative transpiration was used as consumptive use. The slope of cumulative transpiration was derived from Normalized Difference Vegetation Index (NDVI). Interception threshold to estimate effective precipitation was assumed to be 2 mm/day. | Daily transpiration was used as consumptive use. Total evaporation estimate came from the annual water balance (i.e. $P$ and $Q$ data), interception was estimated through model simulation, and daily transpiration was downscaled from long term average transpiration using estimates of daily potential transpiration. | Total time-variable actual evaporation, which includes all evaporation components (e.g. transpiration, interception), was determined from the surface energy balance (not from a water balance). Interception estimation is no longer needed, as it is implicit in both $P$ and $E$ data, and therefore cancels out. Further, we included the effect of evaporation from irrigation. |
| **Mass balance algorithm execution** | Mass Curve Technique | Daily water balance model with interception and root zone storage reservoir. Deficit increases when transpiration exceeds effective precipitation. Any excess precipitation is assumed to runoff directly. | Daily water balance with root zone storage capacity reservoir. Deficit $D$ increases when total evaporation exceeds total precipitation, and decreases when $P>E$ and $D>0$. Excess precipitation is assumed to be runoff or recharge. |
| **Identification of the most suitable drought return period** | Identified through the best runoff simulation performance across the catchment. | Assumed 10 years across the catchment based on Gao et al. (2014). | Differentiated drought return periods are identified for different land use types using evaporation simulation performance. |

**Table 2.** $\epsilon_{\text{RMS}}$ and $\epsilon_{\text{RMS}}$ improvements in evaporation simulation with $E_{\text{SM}}$ and LandFlux-EVAL as benchmark respectively.

| Monthly $E$ compared | $\epsilon_{\text{RMS}}$ with $E_{\text{SM}}$ as benchmark (mm/year) | | | $\epsilon_{\text{RMS}}$ with LandFlux-EVAL diagnostic as benchmark (mm/year) | | |
|---|---|---|---|---|---|---|
| | Look-up | $S_{\text{R,CRU-SM}}$ | $\epsilon_{\text{RMS}}$ improvement | Look-up | $S_{\text{R,CRU-SM}}$ | $\epsilon_{\text{RMS}}$ improvement |
| Mean | 234 | 230 | 4 | 136 | 126 | 10 |
| Max | 323 | 320 | 3 | 244 | 222 | 22 |
| Min | 189 | 181 | 8 | 143 | 129 | 14 |

---

## Author Comment (AC2) · 22 Feb 2016

We thank Referee #2 for constructive comments and positive feedback. We would here like to briefly respond to the general comments and address the specific comments in a later response. Below, Referee #2's general comments are in bold and our responses are in upright font. We refer to the manuscript for explanations of variables and abbreviations.

**The beginning of Sect. 2.1 should be revised. It refers to a figure which is (for me) not really self-explaining. Also, I had problems while reading Page 9, line 10 "$P$ and the evaporation originating form irrigation". I asked myself why is this evaporation not included in the $F_{\text{out}}$ term. I suggest to write something like "the amount of effective irrigation water (that is evapotranspired by the crops)".**

We will improve the explanation of the method in the beginning of Sect. 2.1.

Whether we include it in the $F_{\text{in}}$ or $F_{\text{out}}$ term is not relevant for the final outcome. Since irrigation is an input to the soil moisture stock, we considered it more appropriate to include it in the inflow. On the other hand, one could also argue that the irrigated water could be subtracted from the outgoing flux, so as to compensate for the additional evaporation. From a water balance point of view, however, it is more logical to consider evaporated water as additional water that was brought into the root zone.

The irrigation evaporation does not only include that transpired by crops, but also the incremental evaporation from surface, wet soil, and ponding water at the tail end of irrigation borders. We will add the term "effective irrigation water" to the explanation for clarity.

**At Section 4.3 the authors assess the improvement of the new root zone storage capacity information within the STEAM model by comparing it to the same product with which the root zone storage capacity was developed. I wonder if this is an independent benchmark product or if you should use $E_{\text{CSM}}$ instead (or only $E_{\text{C}}$) as benchmark. Sure, there is a lack of real observation based benchmark products for evaporation, but this is a weak point. You should select a different benchmark or rephrase the term benchmark.**

We consider validation using $E_{\text{SM}}$ to be appropriate, since the algorithms for estimating $S_{\text{R}}$, and for estimating $E$ in STEAM are very different. First, $S_{\text{R}}$ is derived based on the $E$ overshoot over $P$, whereas STEAM is a process-based model where evaporation originates from five different compartments, each constrained by potential evaporation and related stress functions. This means that it is impossible to reproduce $E_{\text{SM}}$ simply by inserting $S_{\text{R}}$ to STEAM. If $S_{\text{R}}$ is zero because $E_{\text{SM}}$ never overshoots precipitation, STEAM soil evaporation and transpiration would become zero. If extreme $S_{\text{R}}$ are produced because $E_{\text{SM}}$ is unrealistically large, STEAM evaporation will not approach $E_{\text{SM}}$, since it will be capped by potential evaporation. Second, consider

also that the precipitation products (CRU and CHIRPS respectively) used for deriving $S_R$ differ from the precipitation forcing (ERA-Interim) used in STEAM. Third, $E_{SM}$ and STEAM are truly independent to each other as well. Whereas STEAM is process and water balance based, the ensemble $E$ product is based on a combination of two($E_{SM}$)/three($E_{CSM}$) well established energy balance methods. The only difference of the new STEAM simulations is the inclusion of updated information on root zone storage so that during longer periods of drought, more realistic estimations of continued evaporation processes can be expected. Last, $S_{R,CRU-SM}$ is based on a single year value of $E_{SM}$ (i.e., the year of maximum storage deficit), whereas the analyses of improvements were based on the entire available time series of 10-11 years. Thus, the fact that $S_{R,CRU-SM}$ dimensioned on one year of $E_{SM}$ nevertheless improves $E$ simulation in STEAM with regard to 10-11 years of $E_{SM}$ (i.e., the overall $\epsilon_{RMS}$ decreases when $S_{R,CRU-SM}$ is used in STEAM) is a strong indication that the storage capacity correction was implemented for the right reason. We maintain that the comparison with $E_{SM}$ is useful and will clarify our arguments in the revised manuscript. Note also that STEAM is not calibrated by any means.

The referee suggests us to use $E_{CSM}$ (or only CMRSET) as benchmark. Figure S4 (in the Supplementary Information) already shows the root mean square error ($\epsilon_{RMS}$) improvements by latitudinal bands when using $E_{CSM}$ as benchmark. Since $E_{CSM}$ does not have global coverage, it was not possible to use it for the climate and land cover based analyses in Sect 4.3 and 4.4. In Figure 1 below, we also include $\epsilon_{RMS}$ improvements by latitudinal bands for the individual evaporation products CMRSET ($E_C$), SSEBop ($E_S$), and MODIS16 ($E_M$) when comparing monthly evaporation for the years 2003-2012. The use of more than one evaporation dataset decreases uncertainties related to individual evaporation products because there is simply not one single preferred model. Research executed by Hofste (2014) for the Nile basin demonstrated that the performance of an ensemble $E$ product is significantly better than using individual $E$ products, something that was confirmed by Simons et al. (in review) in Vietnam. Thus, we maintain that it is more useful to use an ensemble evaporation product as

**Table 1.** $\epsilon_{RMS}$ and $\epsilon_{RMS}$ improvements in evaporation simulation with $E_{SM}$ and LandFlux-EVAL as benchmark respectively.

| Monthly $E$ compared | $\epsilon_{RMS}$ with $E_{SM}$ as benchmark (mm/year) | | | $\epsilon_{RMS}$ with LandFlux-EVAL diagnostic as benchmark (mm/year) | | |
|---|---|---|---|---|---|---|
| | Look-up | $S_{R,CRU-SM}$ | $\epsilon_{RMS}$ improvement | Look-up | $S_{R,CRU-SM}$ | $\epsilon_{RMS}$ improvement |
| Mean | 234 | 230 | 4 | 136 | 126 | 10 |
| Max | 323 | 320 | 3 | 244 | 222 | 22 |
| Min | 189 | 181 | 8 | 143 | 129 | 14 |

benchmark reference.

To address the referee's concern of interdependency, we cross-check the mean monthly STEAM evaporation based on $S_{R,CRU-SM}$ (2003-2013) with the mean monthly LandFlux-EVAL diagnostic ensemble evaporation (1989-2005) (Mueller et al., 2013), see comparison in Table 1 above. It appears that the $\epsilon_{RMS}$ improvements are even greater (mean improvement 10 mm/year instead of 4 mm/year), but with the greatest improvements in maximum monthly evaporation instead of minimum monthly evaporation. The LandFlux-EVAL diagnostic product include the evaporation products: PRUNI, MPIBGC, CSIRO, GLEAM, and AWB. Since this product includes GLEAM, which relies on water balance calculations and soil layer depth assumptions, we consider the use of this product inappropriate for our purposes and would refrain from including this comparison in the manuscript.

**Most of the figures are very tiny, and sometimes due to the choice of color very hard to read (e.g. Fig. A1). Please take care of figure size in the final production phase of the manuscript if it is accepted.**

Thank you for pointing this out. We will improve the choice of colours and increase the figure sizes where needed.
**References**

Hofste, R.W., 2014. Comparative Analysis Among Near-Operational Evapotranspiration Products for the Nile Basin Based on Earth Observations, M.Sc. thesis, Faculty of Civil Engineering and Geosciences, Delft University of Technology: 24 pp

Mueller, B., Hirschi, M., Jimenez, C., Ciais, P., Dirmeyer, P. A., Dolman, A. J., ... Seneviratne, S. I. (2013). Benchmark products for land evapotranspiration: LandFlux-EVAL multi-data set synthesis. Hydrology and Earth System Sciences, 17(10), 3707–3720. http://doi.org/10.5194/hess-17-3707-2013

Simons, G.W.H., W.G.M. Bastiaanssen, L.A. Ngô, C.R. Hain, M.C. Anderson and G. Senay, 2016. Integrating open-access satellite-derived global data products as a pre-analysis for hydrological modelling studies: a case study for the Red River Basin, MDPI Remote Sensing (in review)

[Figure]

[Figure]

**Fig. 1.** Root mean square error improvements in STEAM evaporation simulation based on ${{S}_\textrm{R,CRU-SM}}$ when using various E data as benchmark reference.

---

## Author Comment (AC3) · 30 Mar 2016

We are grateful for the constructive comments, and would here like to respond to Referee #1's specific comments. Below, referee #1's specific comments are in bold and our responses are in upright font. We refer to the manuscript for explanations of variables and abbreviations.

**P10, L13: "The resetting of this limited number of pixels." Please specify what is the percentage number of pixels for which resetting was needed.**

This is an error that slipped through. The resetting was first introduced when we allowed accumulation to persist for two years, but we later changed this threshold to three years. In fact, we do not reset any grid cells when the threshold of persistent $D$ accumulation years is set to three years. We will delete the sentences: "In addition, $D$

is reset to zero by the end of a three years period in a few grid cells where $D$ accumulation persist for three years or more. Such increases are likely the effect of lateral supply of water, or reflect erroneous combinations of $P$ and $E$. The resetting of this limited number of pixels does not affect the outcome of this study in any measureable way."

**P10, L19-23: Simply say that $S_R$ is the maximum of the obtained $D$ values.**

We replace

"Finally, in addition to the moisture deficits with a specific probability of exceedance, we also define the largest value of the moisture deficits $D$ over the considered time series of observation, which, assuming the ecosystem was able to deal with this deficit, would be the estimate of the root zone storage capacity ($S_R$):"

with

"Finally, the root zone storage capacity $S_R$ is defined as the maximum of the obtained $D$ values:"

**P11, L8-11: This paragraph is not clear to me, please revise.**

We replace:

"During wet spells, additional fluxes from the soil system include surface runoff and drainage into groundwater. These fluxes only occur after certain levels of saturation have been achieved. Therefore, during prolonged dry spells, which are critical for sizing the root zone storage requirement, these fluxes may be neglected."

with

"During dry periods, the magnitude of surface runoff and deep drainage is usually small, and therefore is assumed to not affect root zone storage capacity calculations."

**P12, L10: The $C$ parameter values is set to a value equal to 0.1. Why? What is the**
**impact on the results? Why "$C$" in equation (7) is different from "$c$" in equation (6)?**

$C$ is in the unit of m and $c$ is dimensionless. However, we have decided to change the soil moisture stress function in STEAM in order to remove the arbitrariness of picking a parameter. Instead of the soil moisture stress function taken from (Matsumoto et al., 2008), we now use a soil moisture stress function that takes the shape of (van Genuchten, 1980)'s function for dimensionless water content:

$$f(S) = \frac{S}{S_R} \tag{1}$$

We add a comparison to the Supplement showing the differences between STEAM using the Matsumoto function and the van Genuchten function with root zone storage capacity.

**P12, L18: $S_{R,CRU\text{-}SM}$ is not defined, only later in the text.**

Since $\epsilon_{RMS}$ improvements are measured also for other variables than the ones exemplified in Eq. 8, we will for clarity replace the example variables $E_{SM}$ and $S_{R,CRU\text{-}SM}$ with the generic variable names $E_{benchmark}$ and $S_{R,new}$. These variable names will be explained in the text that explain the equation.

**P12, L20: The formulation of equation (8) is wrong for me. The root mean square error should be between $E_{SR,STEAM}$ and $E_{SM}$, not only for $E_{SR,STEAM}$ or $E_{SM}$. Please reformulate.**

Thank you for pointing this out, we correct Eq. 8 to:

$$\varepsilon_{RMS,imp} = \left[\varepsilon_{RMS}(E_{S_{R,STEAM}}, E_{benchmark})\right] - \left[\varepsilon_{RMS}(E_{S_{R,new}}, E_{benchmark})\right]$$

(2)

.

**P13, L16-17 The symbols $\sigma$ and $\sigma_{S_{\text{R,CRU-SM}}}$ are missing in the text. Please correct.**

We will correct this.

**P16, L13: ERA-I evaporation is used as forcing of STEAM. What is the output of STEAM? Is it the actual evaporation? It should be clarified and clearly distinguished from potential evapotranspiration throughout the text.**

The output of STEAM is actual evaporation. ERA-I evaporation was only used to scale calculated daily values of potential evaporation to 3 hours resolution. We will reformulate as follows:

"Input ERA-I data to STEAM were at 3 h and 1.5 degree resolution and include: precipitation, snowfall, snowmelt, temperature at 2m height, dew point temperature at 2 m height, wind speed vector fields (zonal and meridional components) at 10 m height, incoming shortwave radiation, net long-wave radiation, and evaporation (only used to scale potential evaporation from daily to 3 h)."

**P16, L27-28: The methods used for downscaling/upscaling the different datasets should be described.**

For greater clarity, we will reformulate the following:

"Data with other resolutions than 0.5 have been either upscaled by averaging or downscaled by grid cell values transferring."

as:

"Data with finer resolution than 0.5 have been upscaled to 0.5 by simple averaging (i.e., assuming that the value of a 0.5 grid cell correspond to the mean of the overlapping finer grid cell values). Data with coarser resolution than 0.5 were downscaled by oversampling (i.e., transferring grid cell values assuming that the finer 0.5 grid cell values

correspond to those overlapped by the coarser degree grid cell values)."

**P17, L12: "$S_R$ estimated are generally larger", larger than? Please clarify.**

It should read "generally large". We will correct this in the revised manuscript.

**P19, L22-P20, L8: Too many details are given here for the description of the differences of the simulated evaporation data. It is difficult to follow, please reduce the text focusing on the most relevant differences.**

We reformulate the following at P19, L22-P20, L8:

"Figure 6 compares the STEAM-simulated evaporation when using, on the one hand, $S_{R,CRU-SM}$ and, on the other, the look-up table based $S_{R,STEAM}$. The effects on evaporation vary with geography and season. The differences are mainly found in South America outside the tropical wet rainforests, in the Sahel, south of the Congo rainforests and in parts of Southeast Asia. January evaporation simulated with $S_{R,CRU-SM}$ is lower in particularly south of the Sahara, Central America, India, and Southeast Asia, and higher in Argentina. April evaporation shows only local increases in Central America, Sahel, and Southeast Asia, and minor decreases in South Africa, China, and Argentina. July evaporation shows the largest differences, with both strong evaporation reductions in Brazil, Canada and Europe, and significant increases in the seasonally dry tropical forests in Brazil and in Central Africa. In October, changes in evaporation are again less widespread and mainly affecting South America. It appears that $S_{R,CRU-SM}$ has the greatest potential to influence model simulations for the hot and dry seasons, and for the seasonal tropical forests where the root zone storage capacity varies strongly."

to:

"Figure 6 compares the STEAM-simulated evaporation when using, on the one hand, $S_{R,CRU-SM}$ and, on the other, the look-up table based $S_{R,STEAM}$. In general, $S_{R,CRU-SM}$ estimated higher evaporation rates in the tropics and lower evaporation

in the subtropics and temperate zone. In particular, the differences are pronounced during the warm and dry seasons. For example, the evaporation reductions with $S_{\mathrm{R,CRU\text{-}SM}}$ is widespread in the Northern Hemisphere during its summer month July. During the dry seasons (e.g., January in the Sahel, July in Congo south of the Equator), the evaporation increase is the most significant. Moreover, the change in evaporation also depend on land cover type. In South America, evaporation increases in the seasonal tropical forests of the Amazon, whereas evaporation decreases in the savannas and shrublands in the south. These results suggest that $S_{\mathrm{R,CRU\text{-}SM}}$ has the greatest potential to influence model simulations for the hot and dry seasons, in regions where the root zone storage varies strongly."

**P22, L20: Recent studies have obtained huge differences between global scale precipitation datasets (e.g., Trenberth et al. (2014), Herold et al. (2016)). It seems not true that evaporation data (on a global scale) have larger spread than precipitation data. Please reformulate.**

We reformulate the paragraph starting at L.17 as follows: "Finally, the quality of the estimated $S_{\mathrm{R}}$ is dependent on the quality of the input evaporation and precipitation data. In this study, the choice of remotely sensed evaporation products influenced the resulting $S_{\mathrm{R}}$ more than the choice of precipitation product, see the Supplement. In particular, the largest standard deviations in the ensemble evaporation products are located in central South America, the Sahel, India, and northern Australia (see Fig. 2e, 2f). To reduce uncertainty, the presented method is preferably applied using ensemble products based on reliable evaporation and precipitation datasets identified in comparison and evaluation studies (e.g., Bitew Gebremichael, 2011; Herold, Alexander, Donat, Contractor, Becker, 2015; Hessels, 2015; Hofste, 2014; Hu, Jia, Menenti, 2015; Moazami, Golian, Kavianpour, Hong, 2013; Trambauer et al., 2014; Trenberth et al., 2013; Yilmaz et al., 2014)"

**P25, L13: It seems that "and $S_{\mathrm{R,CRU\text{-}SM}}$ is missing here.**

True, we will correct this.

**Tables/Figures: Please check captions for symbols. Captions should be selfde-scribing.**

We will describe all symbols in the captions.

**1 References**

Bitew, M. M. and Gebremichael, M.: Evaluation of satellite rainfall products through hydrologic simulation in a fully distributed hydrologic model, Water Resour. Res., 47(6), W06526, doi:10.1029/2010WR009917, 2011.

van Genuchten, M. T.: A Closed-form Equation for Predicting the Hydraulic Conductivity of Unsaturated Soils, Soil Sci. Soc. Am. J., 44, 892–898, 1980. Herold, N., Alexander, L. V., Donat, M. G., Contractor, S. and Becker, A.: How much does it rain over land?, Geophys. Res. Lett., 43(1), n/a–n/a, doi:10.1002/2015GL066615, 2015.

Hessels, T. M.: Comparison and Validation of Several Open Access Remotely Sensed Rainfall Products for the Nile Basin, Delft University of Technology., 2015.

Hofste, R. W.: Comparative analysis of near-operational evapotranspiration products for the Nile basin based on Earth Observations; First steps towards an ensemble ET product, Delft University of Technology. [online] Available from: http://repository.tudelft.nl/assets/uuid:16659a39-3256-4ff9-9930-81ac4dfb4018/maindoc.pdf, 2014.

Hu, G., Jia, L. and Menenti, M.: Comparison of MOD16 and LSA-SAF MSG evapotranspiration products over Europe for 2011, Remote Sens. Environ., 156, 510–526, doi:10.1016/j.rse.2014.10.017, 2015.

Matsumoto, K., Ohta, T., Nakai, T., Kuwada, T., Daikoku, K., Iida, S., Yabuki, H.,

Kononov, A. V., van der Molen, M. K., Kodama, Y., Maximov, T. C., Dolman, A. J. J. and Hattori, S.: Responses of surface conductance to forest environments in the Far East, Agric. For. Meteorol., 148(12), 1926–1940, doi:10.1016/j.agrformet.2008.09.009, 2008.

Moazami, S., Golian, S., Kavianpour, M. R. and Hong, Y.: Comparison of PERSIANN and V7 TRMM Multi-satellite Precipitation Analysis (TMPA) products with rain gauge data over Iran, Int. J. Remote Sens. [online] Available from: http://www.tandfonline.com/doi/abs/10.1080/01431161.2013.833360 (Accessed 19 February 2016), 2013.

Trambauer, P., Dutra, E., Maskey, S., Werner, M., Pappenberger, F., van Beek, L. P. H. and Uhlenbrook, S.: Comparison of different evaporation estimates over the African continent, Hydrol. Earth Syst. Sci., 18(1), 193–212, doi:10.5194/hess-18-193-2014, 2014.

Trenberth, K. E., Dai, A., van der Schrier, G., Jones, P. D., Barichivich, J., Briffa, K. R. and Sheffield, J.: Global warming and changes in drought, Nat. Clim. Chang., 4(1), 17–22, doi:10.1038/nclimate2067, 2013.

Yilmaz, M. T., Anderson, M. C., Zaitchik, B., Hain, C. R., Crow, W. T., Ozdogan, M., Chun, J. A. and Evans, J.: Comparison of prognostic and diagnostic surface flux modeling approaches over the Nile River basin, Water Resour. Res., 50(1), 386–408, doi:10.1002/2013WR014194, 2014.

---

## Author Comment (AC4) · 30 Mar 2016

We are grateful for the constructive comments, and would here like to briefly respond to Referee #2's specific comments. Below, referee #2's specific comments are in bold and our responses are in upright font. We refer to the manuscript for explanations of variables and abbreviations.

**I suggest changing the first word of the title to Global-scale. That is more reflecting the issue, that the product is done for global land surface (but one can also live with just "global")**

"Global" is succinct and also common in other similar context, e.g., "Global Soil Map" and "Global Land Cover". For these reasons, we intend to keep "global" in the title.

[Figure]

**Page 6, line 23: I would recommend a new section starting with "For global" or revising the first sentence. The last sentence of the section are general statements of model calibration, but to my knowledge, none of the models described here is calibrated directly on root zone storage capacity as it is written in the first sentence. Please revise.**

It's true that not all of the mentioned studies calibrated directly on root zone storage capacity. Instead, they calibrated various both physically-based and conceptual parameters (sometimes related to root zone storage capacity). We will replace the sentence at P6 L23-27:

"For global hydrological models, calibration has mostly been performed separately for a selection of large river basins and transferred to other regions using a regionalisation approach (Güntner, 2008; Hanasaki et al., 2008; Hunger and Döll, 2008; Nijssen et al., 2001; Werth and Güntner, 2010; Widén-Nilsson et al., 2007)."

with the following:

"For global hydrological models, parameters can be calibrated separately for a selection of gauged river basins and transferred to neighbouring ungauged catchments (Döll et al., 2003; Güntner, 2008; Hunger and Döll, 2008; Nijssen et al., 2001; Widén-Nilsson et al., 2007) . This procedure, known as regionalisation, has (to our knowledge) only been performed for other parameter values than the root zone storage capacity, although the principle does not change with the parameters tuned."

**Page 8, line 2: you should change "stores" to "storages" as this is more common term in that sense**

We will change "stores" to "storages" here.

**Page 8, line 10: change "global" by "global-scale"**

"Global" and "global-scale" are both grammatically correct here. Since we kept "global" in the title, we intend to also stick to "global" in this phrase.

**Page 9, line 15 f: strange sentence. Why should one understand that irrigation is included in precipitation data? Or do you refer to satellite precipitation data? This sentence needs to be revised**

At L15, it reads "This is because irrigation is captured in satellite-based evaporation data, but obviously not in precipitation data". We never claimed that irrigation is included in the precipitation data. It is part of the incoming flux, but not part of the precipitation. Therefore we are not sure what the referee means by "Why should one understand that irrigation is included in precipitation data?".

**Page 10, line 13: please be precise and present the number of pixels that are affected.**

The statement on $D$ reset is an error that slipped through. The resetting was first introduced when we allowed accumulation to persist for two years, but we later changed this threshold to three years. In fact, we do not reset any grid cells when the threshold of persistent $D$ accumulation years is set to three years. We will delete the sentences "In addition, $D$ is reset to zero by the end of a three years period in a few grid cells where $D$ accumulation persist for three years or more. Such increases are likely the effect of lateral supply of water, or reflect erroneous combinations of $P$ and $E$. The resetting of this limited number of pixels does not affect the outcome of this study in any measureable way."

**Page 11, line 21: a dot at end of section is missing.**

We will correct this.

**Page 13,line 3: I wonder how many of the 1.5 deg grid cells are available for each land cover class, if land cover needs to be at least 90 % of a single land cover. Maybe I have misinterpreted the information, but is it correct that you used MODIS land cover with 0.05 resolution to assess land cover for the 1.5 cell? So, the 1.5 cell consists of 90 0.05 tiles, and at least 81 of them needs to be**

in one land cover class. The global pattern of land cover is very heterogeneous and think it is important for interpreting the results if you write (e.g. in a table) the number of grid cells per land cover class that went into that analysis.

We will provide information on the number of grid cells per land cover class in the revised manuscript.

**Page 15, line 8 ff: please describe at least in some sentences the temporal downscaling of monthly evaporation and precipitation data.**

We downscaled monthly evaporation and precipitation by applying the daily-to-monthly ratios derived from ERA-Interim $E$ and $P$ data. We will added the following:

"In the temporal downscaling, we first established the ratios between daily values to the mean monthly ERA-Interim, and second, used the relationship to estimate daily values from monthly $E_{\mathrm{SM}}$ or $E_{\mathrm{CSM}}$ values."

**Page 16, line 24: again, it would be nice to have an idea, how many grid cells are used per land cover class.**

We will add this to the revised manuscript.

**Page 20, line 14 f: Hard to judge that because de Boer-Euser is not available for the reviewers. Maybe you should write in a few sentences what is written there.**

De Boer-Euser et. al. is now published. We will update the reference.

**Page 21, line 14 f: The term "speculate" is not nice in a paper. Is there any reference for the message "deciduous forests need a large root zone storage capacity to cater for dry periods during their most active summer months" available or is it something you found out during your research? Please try to verify this. Same for the next sentence.**

The coping mechanism of deciduous forest is to shed leaves when there is too much moisture stress. In semi-arid tropical climates we can see this happen when the wet

season comes to an early end; likewise we see this in temperate climates when the cold season starts early. However, shedding the leaves during the wet season (semi-arid tropics) or the growing season (summer in temperate climates) is not attractive because it prevents reproduction. So the only option to prevent lack of reproduction is to create adequate storage. This storage needs to be big enough so as to reach reproductive age.

Therefore, it also depends on the age of the species. An annual species only needs to cater for an average year (or somewhat drier than average year), particularly if there is a seed stock that can survive drought, or if the plant can go dormant (grasses). So age also comes into the equation. In the analysis, the "deciduous forest" constitute of both "deciduous needleleaf forest (DNF)" and "deciduous broadleaf forest (DBF)". We lumped the needleleaf and broadleaf to prevent statistical artefact due to a low number of grid cells covered (i.e., only 13 grid cells of deciduous broadleaf forest and 31 grid cells of DNF, see figure below). However, this concealed an effect that probably is real – i.e., that DNF performs the best with a very long return period, whereas DBF performs the best with a shorter return period.

It appears that the age structure between DNF and DBF are radically different. Global mapping of forest age structure using country database and MODIS data (Poulter, 2012) show that DNF have a very high age, whereas large areas of DBF are notoriously young. The majority of the forests were > 80 years in the region (i.e. Siberia) with highest concentration of DNF (i.e., where our analysis covered due to the 90 % single land use occupancy requirement). By contrast, 1-30 years and 31-80 years were far more common in the regions with high DBF concentrations (i.e., in Easter U.S. and South America). Based on U.S. forest inventories, (Hicke et al., 2007) report mean forest age of 60-100 years, with hardwood (incl. DBF) younger than softwood (incl. DNF). In terms of longevity, it also seems that gymnosperms (incl. DNF) beat the angiosperms (incl. DBF). The mean maximum age of 60 species of reported North American angiospermous trees is 248 years, whereas it is 596 years for the 50 species

of gymnospermous trees (Larson, 2001; Loehle, 1988). Longevity could be explained by strong defence mechanisms against fungi and insects, lack of physical environmental damage, but also low occurrence of environmental stress such as drought (Larson, 2001). Thus, we think it could be correct that deciduous needleleaf forests, which is older, also have root zone storage capacities that have catered for longer drought return periods.

In the revised manuscript, we will discard the lumping of needleleaf and broadleaf, and instead report all forest types separately together with grid cell numbers. We will delete the wording "speculate" and the possible explanation of poor snow simulation in STEAM. Instead, we will explain the result by relating it to studies of forest age.

(We changed the soil moisture stress function in STEAM, see response to referee #1's specific comments and Axel Kleidon. This changed the best return period for mixed forest, woody savannah, savannah and evergreen broadleaf forest, but does not affect the deciduous forests and the discussion above.)

**Page 23, line 2: after "from" can be misinterpreted and is not complete. I suggest to write: "...from remotely sensed evaporation, remotely sensed and station based precipitation and model based irrigation**. . ."

We agree it's not complete, but the full sentence is also cumbersome to read. In principal, the method used in the paper is for use with remotely sensed data. Irrigation only affects the irrigated regions in the world. We will revise to the following:

"This study presents a method to estimate root zone storage capacity in principle from remotely sensed evaporation and observation-based precipitation data, by assuming that plants do not invest more in their roots than necessary to bridge a dry period."

**Page 23, line 7: did you compare $S_{R,CHIRPS-CSM}$ with $E_{SM}$ or with $E_{CSM}$? Is that conclusion somewhere covered in the paper?**

Yes, see Fig. S4 in the Supplementary Information. The results are discussed in the

Supplementary Information Sect. 1.3.

**References, general: why are the page numbers written after the reference? Is this the new standard of HESS?**

We will ask the typesetter at Copernicus Publications for the re-submission.

**Page 28, line 12: belongs "Open Access" really to the journal title?**

The journal is Remote Sensing. We will correct this reference.

**Page 31, line 3: Please check the citation. It is a master thesis, and I am not sure if there are so many co-authors.**

The committee members were erroneously listed as co-authors. We will correct this in the revised manuscript.

**Page 31, line 22: Check names**

We will correct the capitalising of author names.

**Page 32, line 4: does Jennings really have CMHCMH as initials? I tried to get access but that failed. Could you maybe update the resource?**

It should be CMH once. We will correct this in the revised manuscript.

**Page 33, line 24: soil should be in lower case**

We will correct this.

**Page 35, line 8: check initials from last co-author**

We will correct the last co-author initials to Xu, C-Y.

**Page 40, Fig 3: For Africans desert region, the CRU-SM product has obviously values of 0, whereas the CHIRPS-CSM product is >0, and if I see it correctly at Fig 3c, it has reasonable large values. Could you explain somewhere, where the difference comes from - is it due to precip product or due to additional evaporation**

**product?**

This comes from the CMRSET evaporation product. We will add figures showing the $S_R$ as combination of the separate $E$ and $P$ products in the Supplementary Information, and this will become apparent. Also added an explanation in Sect. 4.3:

"The positive values of $S_{R,\text{CHIRPS}-\text{CSM}}$ in the Sahara desert are caused by overestimation of evaporation in the CMRSET evaporation product, (see also the Supplement)."

**Page 44, Fig 7: please write after aridity index that the calculation can be found in Sect B1.**

We will add this.

**1 References**

Döll, P., Kaspar, F. and Lehner, B.: A global hydrological model for deriving water availability indicators: model tuning and validation, J. Hydrol., 270(1-2), 105–134, doi:10.1016/S0022-1694(02)00283-4, 2003.

Güntner, A.: Improvement of global hydrological models using GRACE data, Surv. Geophys., 29(4-5), 375–397, doi:10.1007/s10712-008-9038-y, 2008.

Hicke, J. A., Jenkins, J. C., Ojima, D. S. and Ducey, M.: SPATIAL PATTERNS OF FOREST CHARACTERISTICS IN THE WESTERN UNITED STATES DERIVED FROM INVENTORIES, Ecol. Appl., 17(8), 2387–2402, doi:10.1890/06-1951.1, 2007.

Hunger, M. and Döll, P.: Value of river discharge data for global-scale hydrological modeling, Hydrol. Earth Syst. Sci., 12(3), 841–861, doi:10.5194/hess-12-841-2008, 2008.

Larson, D. W.: The paradox of great longevity in a short-lived tree species, Exp. Gerontol., 36(4-6), 651–673, doi:10.1016/S0531-5565(00)00233-3, 2001.

Loehle, C.: Tree life history strategies: the role of defenses, Can. J. For. Res., 18(2), 209–222, doi:10.1139/x88-032, 1988.

Nijssen, B., O'Donnell, G. M., Lettenmaier, D. P., Lohmann, D. and Wood, E. F.: Predicting the discharge of global rivers, J. Clim., 14(15), 3307–3323, doi:10.1175/1520-0442(2001)014<3307:PTDOGR>2.0.CO;2, 2001.

Poulter, B.: Forest age datasets, AGU, 12 [online] Available from: http://www.nacarbon.org/meeting_ab_presentations/2013/poulter_nacp_2013a.pdf (Accessed 1 March 2016), 2012.

Widén-Nilsson, E., Halldin, S. and Xu, C.-Y.: Global water-balance modelling with WASMOD-M: Parameter estimation and regionalisation, J. Hydrol., 340(1-2), 105–118, doi:10.1016/j.jhydrol.2007.04.002, 2007.
* * *
[Figure]

**Mean $\epsilon_{RMS}$ improvements by land cover type**

Legend:
- $S_{R,CRU\text{-}SM,2yrs}$
- $S_{R,CRU\text{-}SM,5yrs}$
- $S_{R,CRU\text{-}SM,10yrs}$
- $S_{R,CRU\text{-}SM,20yrs}$
- $S_{R,CRU\text{-}SM,40yrs}$
- $S_{R,CRU\text{-}SM,60yrs}$

Y-axis: Mean $\epsilon_{RMS}$ improvement, (mm/year)

X-axis categories: Evergr. needlel. forest (9), Evergr. broadl. forest (239), Dec. needlel. forest (31), Dec. broadl. forest (13), Mixed forest (95), Open shrubland (568), Woody savannah (85), Savannah (133), Grassland (314), Cropland (123), Barren land (554)

X-axis label: $S_{R,CRU\text{-}SM}$ dataset, (-)

**Fig. 1.** The mean RMSE improvement in simulated monthly E (2003–2013) by implementing SR,CRU−SM,Lyrs instead of SR,STEAM in STEAM. The number of represented grid cells are in parenthesis.

---

## Author Comment (AC5) · 30 Mar 2016

We are grateful for the constructive comments and shared insights, and would here like to respond to Axel Kleidon's comments. Below, Axel Kleidon's comments are in bold and our responses are in upright font. We refer to the manuscript for explanations of variable names and abbreviations.

**Page 4-6: The introduction contains a nice review of the different methodologies. I think for what you describe as the "root distribution modelling approach" is more appropriately labeled an "optimization/maximization approach", as it infers rooting properties from some ecological cost function. Also, Kleidon and Heimann 1998 did not use an inverse approach, but an optimization approach, even though in a highly simplified way and without the use of an explicit root**

**distribution, so it may be better to refer to such a class of "optimization" approaches.**

This is a great suggestion. We will re-label the "root distribution modelling approach" as "optimisation approach". We will also move the reference Kleidon and Heimann 1998 to the "optimisation approach" category.

**page 7, line 10: I think more relevant is here a link to cost-benefit type of analysis rather than evolution. It may be appropriate to refer to the classic book edited by Givnish "On the economy of plant form and functioning" by Cambridge University Press.**

Gao et al., (2014) emphasised the role of co-evolution of climate, ecosystems, and hydrology. In a way, cost-benefit strategy of plants is a result of evolution. Thank you for the reference suggestion. It is a seminal book, which we will refer to here. We will replace the original statement:

"Their results suggested that ecosystems develop their root zone storage capacity to deal with droughts with specific return periods, beyond which the costs of carbon allocation to roots are too high from an evolutionary point of view."

with the following:

"Their results suggested that ecosystems develop their root zone storage capacity to deal with droughts with specific return periods, beyond which the costs of carbon allocation to roots are too high from the perspective of the plants. This resonates well with past economic analyses of plant behaviour and traits, e.g. Givnish, (1986)."

**page 10, line 14: "in any measureable way" sounds rather strong. Perhaps better to say that it only affects results to a small extent?**

The statement on $D$ reset is an error that slipped through. The resetting was first introduced when we allowed accumulation to persist for two years, but we later changed this threshold to three years. In fact, we do not reset any grid cells when the threshold

of persistent $D$ accumulation years is set to three years. We will delete the sentences "In addition, $D$ is reset to zero by the end of a three years period in a few grid cells where $D$ accumulation persist for three years or more. Such increases are likely the effect of lateral supply of water, or reflect erroneous combinations of $P$ and $E$. The resetting of this limited number of pixels does not affect the outcome of this study in any measureable way."

**page 12, line 12: It would be nice to see how well the two formulations of the stress function compare to each other. Can you show this in a figure?**

we have decided to change the soil moisture stress function in STEAM in order to remove the arbitrariness of picking a parameter. Instead of the soil moisture stress function taken from (Matsumoto et al., 2008), we now use a soil moisture stress function that takes the shape of (van Genuchten, 1980)'s function for dimensionless water content:

$$f(S) = \frac{S}{S_R} \tag{1}$$

We add a comparison to the Supplement showing the differences between STEAM using the Matsumoto function and the van Genuchten function.

**page 14, line 14: "electro-magnetic spectrum". do you mean different wavelengths/bands?**

Electro-magnetic spectra are described by wavelength, frequency, and photon energy. In this context, both electro-magnetic spectrum and wavelength can be used. Since we in the subsequent sentences refer to regions in the electro-magnetic spectrum (i.e., visible, infrared), we considered it more appropriate to refer to the electro-magnetic spectrum.

**page 16, line 14: "wind speed in two directions" really? If so, why do you use**

**both directions? Or do you use the two measurements to calculate wind speed?**

ERA-Interim wind speed vector fields are provided in two components. We will revise this to the following for clarity: "wind speed vector fields (zonal and meridional components) at 10m height"

**page 18, lines 4-11/Figure 4: the correspondence (or disagreements) between the data sets would be easier to see in a scatterplot, where the different data sets are compared at a grid-by-grid scale. How well they correspond is then reflected by the slope of the regression as well as the r2 value. It is probably not necessary to show all scatterplots (or add them as supplementary), but I think this type of analysis would really help to identify how well the different data sets compare to each other.**

Great suggestion, we will add comparison scatter plots to the Supplementary Information.

**page 18, lines 12-28: In the discussion of the differences, it is also important to note that these datasets may use different climate data sets, particularly precipitation. Also, Kleidon (2004) calculated evaporation in a quite simple way, which also is likely to result in differences. What this means is that the differences may not simply reflect on different ways to infer rooting properties, but there is also a component related to the forcing datasets which is difficult to quantify.**

Agree, we add:

"Nevertheless, different input data were also used in the different studies. Thus, it is difficult to attribute the variations in root zone storage capacity estimates to differences in methods or differences in input data."

**page 25, line 2: Note that for the effect of climate change, it also depends on the ability of vegetation to adapt to altered conditions. This aspect should be mentioned.**

Agree, we add: "...depending on the adaptability of vegetation to altered conditions."

**page 38, Figure 1: This figure nicely illustrates the concept. I think it could be made even better if you show the integrated fluxes of Fin-Fout over time in a separate plot above the panel where you show the bins.**

Good idea. We now show the integrated fluxes over time above the panel of bins in Figure 1, see below.

**page 40, Figure 3: Start the caption more descriptive with something like "Estimates of root zone storage capacity of the ...". You may also want to use the same color scale in panel (c) as in Fig. 4 to facilitate comparison?**

We will adopt the same color scale and revise the caption as follows: "Root zone storage capacity estimates of (a) $S_{\mathrm{R,CHIRPS\text{-}CSM}}\cdots$"

**page 43, Figure 6: I find the differences difficult to see. It may be easier to attribute the differences when you use only a few discrete color values with less than 8 shadings so that one more clearly associate the differences in a region with the values. (also applies to other plots)**

We will use discrete colour values in the revised manuscript.

**1 References**

Gao, H., Hrachowitz, M., Schymanski, S. J., Fenicia, F., Sriwongsitanon, N., Savenije, H. H. G. (2014). Climate controls how ecosystems size the root zone storage capacity at catchment scale. Geophysical Research Letters, 41(22), 7916–7923. http://doi.org/10.1002/2014GL061668

Givnish, T. (1986). On the Economy of Plant Form and Function. (T. Givnish, Ed.). Cambridge University Press. Retrieved from

http://www.cambridge.org/cr/academic/subjects/life-sciences/cell-biology-and-developmental-biology/economy-plant-form-and-function-proceedings-sixth-maria-moors-cabot-symposiumcontentsTabAnchor

Matsumoto, K., Ohta, T., Nakai, T., Kuwada, T., Daikoku, K., Iida, S., … Hattori, S. (2008).  Responses of surface conductance to forest environments in the Far East.  Agricultural and Forest Meteorology, 148(12), 1926–1940. http://doi.org/10.1016/j.agrformet.2008.09.009

van Genuchten, M. T. (1980).  A Closed-form Equation for Predicting the Hydraulic Conductivity of Unsaturated Soils. Soil Science Society of America Journal, 44, 892–898.

[Figure]

**Fig. 1.** Conceptual illustration of the algorithm for calculating the root zone storage capacity.

---

## Author Response (AR2)

Dear editor,

We are pleased that the manuscript is now accepted for publication with minor revisions. We have corrected the manuscript with respect to the points detailed in the referee reports.

The point-to-point list below provides an overview of all revisions, whereas the motivations behind the revisions can be found in the separate responses to referee comments (posted in the interactive discussion). A marked-up version of the manuscript can be found by the end of this document.

Beside the revisions in response to referees, we also decided to modify the soil moisture stress function in STEAM from the original formulation of (Matsumoto et al., 2008) to the simpler formulation of (van Genuchten, 1980) in order to remove the arbitrariness of picking a soil moisture stress parameter. The new soil moisture stress function is:

$$f(S) = \frac{S}{S_R} \, ,$$

where $S$ is actual soil moisture storage (m) and $S_R$ is root zone storage capacity (m). This is an improvement, which changes some of the results regarding the best drought return period in woody savannah (from 2 years to 10 years), savannah (from 2 years to 10 years), evergreen forest (from 10 years to 60 years), and mixed forest (from 10 years to 60 years). We added a comparison to the Supplementary Information showing the differences between STEAM using the Matsumoto function and the van Genuchten function. This change in soil moisture stress function only affects the analyses of the "optimal" drought return periods. Thus, the main results including the estimates of root zone storage capacity as well as the $S_R$ values within the Gumbel distribution are entirely unaffected.

Kind regards,

Lan Wang-Erlandsson with co-authors

**Overview of manuscript revisions**

The list below is ordered as the manuscript sections. The first column is the section heading, the second column is the referee comment leading up or related to a change in the manuscript, the third column is the change, and the fourth column is the referee number or AK for Axel Kleidon (S or G for specific and general comments respectively) or "N/A" in the case when we made changes beside what is recommended by the referees. Referee comments that did not generate changes are listed at the end of the table.

| Applies to | Referee comment | Change | R# |
|---|---|---|---|
| **Abstract** | | Edited return periods for land cover types after changes in results after changing to the van Genuchten equation. | N/A |
| | | | |
| **1. Introduction** | Page 4-6: The introduction contains a nice review of the different methodologies. I think for what you describe as the "root distribution modelling approach" is more appropriately labeled an "optimization/maximization approach", as it infers rooting properties from some ecological cost function. Also, Kleidon and Heimann 1998 did not use an inverse approach, but an optimization approach, even though in a highly simplified way and without the use of an explicit root distribution, so it may be better to refer to such a class of "optimization" approaches. | Changed as suggested. | AK |
| | Page 6, line 23: I would recommend a new section starting with "For global" or revising the first sentence. The last sentence of the section are general statements of model calibration, but to my knowledge, none of the models described here is calibrated directly on root zone storage capacity as it is written in the first sentence. Please revise. | Modified as follows:
"For global hydrological models, parameters can be calibrated separately for a selection of gauged river basins and transferred to neighbouring ungauged catchments (Döll et al., 2003; Güntner, 2008; Hunger and Döll, 2008; Nijssen et al., 2001; Widén-Nilsson et al., 2007). This procedure, known as regionalisation, has (to our knowledge) only been performed for other parameter values than the root zone storage capacity, although the principle does not change with the parameters tuned." | 2S |

| | | | |
|---|---|---|---|
| | page 7, line 10: I think more relevant is here a link to cost-benefit type of analysis rather than evolution. It may be appropriate to refer to the classic book edited by Givnish "On the economy of plant form and functioning" by Cambridge University Press. | Modified as follows:
"Their results suggested that ecosystems develop their root zone storage capacity to deal with droughts with specific return periods, beyond which the costs of carbon allocation to roots are too high from the perspective of the plants. This resonates well with past economic analyses of plant behaviour and traits, e.g. (Givnish, 1986)." | AK |
| | Page 8, line 2: you should change "stores" to "storages" as this is more common term in that sense | Changed. | 2S |
| | A very recent paper with the same topic has been published in Journal of Hydrology by Campos et al. (2016, http://dx.doi.org/10.1016/j.jhydrol.2016.01.023). I suggest mentioning and analysing this study. | Recently, this approach has also been applied at the local scale to approximate the root zone storage capacity by minimising water balance modelled and remote sensing based evaporation \citep{Campos2016}. | 1G |
| **1.2 Research aims** | The description of the method should be improved. Is the method the same as in previous studies (e.g., Gao et al. 2014 GRL)? If yes, it should be clearly acknowledged. Is it different from the paper (under review, not available to reviewers) by Boer-Euser et al.? It should be clear to the reader if the novelty of the papers is on the method or in the satellite dataset used as input. Please clarify. | Added:

"While we make use of the same mass balance principle as applied by Gao et al., (2014) and de Boer-Euser et al., (2016), our algorithm is based on indirect measurements of every unique pixel. Methodologically, in contrast to these two studies, the analyses here are carried out on global gridded data rather than by catchment and use total evaporation instead of interception and transpiration estimates." | 1G |
| | | | |
| **2. Methods** | | | |
| **2.1 Estimating root zone storage capacity** | The beginning of Sect. 2.1 should be revised. It refers to a figure which is (for me) not really self-explaining. | Modified from:
"The algorithm is conceptually illustrated in Fig. 1."
To

"The algorithm is explained in this section and conceptually illustrated in Fig. 1." | 2G |

| | | | |
|---|---|---|---|
| | Also, I had problems while reading Page 9, line 10 "P and the evaporation originating form irrigation". I asked myself why is this evaporation not included in the Fout term. I suggest to write something like "the amount of effective irrigation water (that is evapotranspired by the crops)" | Modified from: "…and the evaporation originating from irrigation Firr (i.e., incremental evaporation)".

to:
"…the effective irrigation water Firr (i.e., incremental evaporation from surface, wet soil, and ponding water at the tail end of irrigation borders that originates from irrigation)" | 2G |
| | "the long term average is added in order to compensate for overestimation of evaporation and underestimation of precipitation". Why? | Modified from "…in order to compensate for overestimation of evaporation and underestimation of precipitation." to: "…in order to compensate for lateral inflow or estimation errors in evaporation or precipitation." | 1G |
| | "in order to take into account of surface runoff, D never becomes negative". Again, why? | Modified from: "However, in order to take into account surface runoff, D never becomes negative (see Fig. 1 at t2)."

To:
"However, D never becomes negative by definition, since it can be considered a running estimate of the root zone storage reservoir size (see Fig. 1 at t2). Not allowing negative D also means that any excess precipitation is assumed to be runoff or recharge." | 1G |
| | P10, L13: "The resetting of this limited number of pixels." Please specify what is the percentage number of pixels for which resetting was needed. (1S)

Page 10, line 13: please be precise and present the number of pixels that are affected. (2S) | Error that slipped through, we do not reset values in the latest version of the algorithm.

"In addition, D is reset to zero by the end of a three years period in a few grid cells where D accumulation persist for three years or more. Such increases are likely the effect of | 1S, 2S, AK |

| | | | |
|---|---|---|---|
| | page 10, line 14: "in any measureable way" sounds rather strong. Perhaps better to say that it only affects results to a small extent? (AK) | lateral supply of water, or reflect erroneous combinations of P and E. The resetting of this limited number of pixels does not affect the outcome of this study in any measureable way." | |
| | P10, L19-23: Simply say that SR is the maximum of the obtained D values. | We replace the sentence "Finally, in addition to the moisture deficits with a specific probability of exceedance, we also define the largest value of the moisture deficits D over the considered time series of observation, which, assuming the ecosystem was able to deal with this deficit, would be the estimate of the root zone storage capacity ($S_R$):" with "Finally, the root zone storage capacity $S_R$ is defined as the maximum of the obtained $D$ values:" | 1S |
| | P11, L8-11: This paragraph is not clear to me, please revise. | We replace the paragraph "During wet spells, additional fluxes from the soil system include surface runoff and drainage into groundwater. These fluxes only occur after certain levels of saturation have been achieved. Therefore, during prolonged dry spells, which are critical for sizing the root zone storage requirement, these fluxes may be neglected." with "Surface runoff and drainage into groundwater are fluxes that occur during wet spells, since they require certain levels of soil moisture saturation. These fluxes may, therefore, be neglected during prolonged dry spells when root zone storage requirements are sized." | 1S |
| **2.2 Implementation in a hydrological model** | P12, L10: The C parameter values is set to a value equal to 0.1. Why? What is the impact on the results? Why "C" in equation (7) is different from "c" in equation (6)? (1S) | Soil moisture stress function is changed and parameter c or C are no longer needed. The paragraph on soil moisture stress function is rewritten. | 1S, AK |

| | | | |
|---|---|---|---|
| | page 12, line 12: It would be nice to see how well the two formulations of the stress function compare to each other. Can you show this in a figure? (AK) | | |
| | P12, L18: SR;CRU⬜SM is not defined, only later in the text. | Changed $S_{R,CRU-SM}$ to $S_{R,new}$ in this section. | 1S |
| | | Changed $E_{SM}$ to $E_{benchmark}$ in this section. | N/A |
| | I found the selection of the application for validating the obtained SR maps not correct. In the paper, it is assessed the improvement in estimating evaporation with the new SR parameterization in STEAM. It is fine for me. The problem is that the same evaporation dataset (ESM) used for computing SR is also used for assessing the improvements due to the new SR parameterization. It is a circular argument that is not good. I suggest performing a different validation test. Why not considering the differences in the runoff prediction with the old and new SR parameterization? It looks to me much more relevant, and a good independent evaluation. (1G)

At Section 4.3 the authors assess the improvement of the new root zone storage capacity information within the STEAM model by comparing it to the same product with which the root zone storage capacity was developed. I wonder if this is an independent benchmark product or if you should use ECSM instead (or only EC) as benchmark. Sure, there is a lack of real observation based benchmark products for evaporation, but this is a weak point. You should select a different benchmark or rephrase the term benchmark. | Added "The remote sensing based ensemble evaporation product $E_{SM}$ (and $E_{CSM}$ in the Supplement) was used as benchmark $E_{benchmark}$. This use may seem circular when $E_{benchmark}$ is used to derive $S_{R,new}$, but is in fact valid due to differences in algorithms, precipitation input data, model types, and time span covered. First, the algorithms for estimating $S_{R,new}$, and for estimating $E$ in STEAM are very different. While $S_{R,new}$ is derived based on the $E$ overshoot over $P$, STEAM is a process-based model where evaporation originates from five different compartments, each constrained by potential evaporation and related stress functions. This means that it is impossible to reproduce $E_{benchmark}$ simply by inserting $S_{R,new}$ to STEAM. Second, the precipitation products (CRU and CHIRPS respectively) used for deriving $S_{R,new}$ differ from the precipitation forcing (ERA-Interim) used in STEAM. Third, $E_{benchmark}$ and STEAM are truly independent to each other as well. Whereas STEAM is process and water balance based, the ensemble $E$ product is based on a combination of two($E_{SM}$) or three($E_{CSM}$) energy balance methods. Last, $S_{R,new}$ is based on a single year value of $E_{benchmark}$ (i.e., the year of maximum storage deficit), whereas the analyses of improvements are based on the entire available time series of 10-11 years. The only difference of the new STEAM simulations is the inclusion of updated information on root zone storage so that during longer periods of drought, more realistic estimations of | 1G, 2G |

| | | | |
|---|---|---|---|
| | | continued evaporation processes can be expected. Thus, if $S_{R,new}$ dimensioned on one year of $E_{benchmark}$ nevertheless improves $E$ simulation in STEAM with regard to 10-11 years of $E_{benchmark}$ (i.e., the overall $\varepsilon_{RMS}$ decreases when $S_{R,new}$ is used in STEAM) is a strong indication that the storage capacity correction was implemented for the right reason." | |
| | | Clarified that there is a bin size restriction of minimum 200 grid cells. | N/A |
| | P12, L20: The formulation of equation (8) is wrong for me. The root mean square error should be between ESR;STEAM and ESM, not only for ESR;STEAM or ESM. Please reformulate. | Equation corrected as suggested. | 1S |
| **2.3 Frequency analysis** | P13, L16-17 The symbols sigman and sigmaSR are missing in the text. Please correct. | Corrected. | 1S |
| | | | |
| **3. Data** | | | |
| **3.1. Evaporation and precipitation input for estimating SR** | In most of the paper, only the SR,CRU-SM dataset is analysed. Why two datasets are considered (CHIRPS and CRU)? The real value of considering also the CHIRPS dataset is not clear to me. Please clarify. | Added: "We present $S_{R,CHIRPS-CSM}$, because $P_{CHIRPS}$ is the lead precipitation product and we can make use of three evaporation datasets. However, $P_{CHIRPS}$ is unfortunately not available at the global scale, and CMRSET is not reliable in high latitudes. Thus, we added the global scale $S_{R,CHIRPS-CSM}$ to this study. This allows for application in global scale models as well as investigations at the global scale (e.g., climate and land cover based analyses)." | 1G |
| | The selection of the input datasets is for me a major issue. Again, it should be clarified why satellite-based data are considered for evaporation and not for precipitation. Why satellite-based datasets for precipitation are not considered (e.g., TMPA, CMORPH, PERSIANN)? | Added: "This study required global coverage data at a grid cell resolution for both evaporation and precipitation. Importantly, these products must not be produced using assumptions on root zone storage capacity, to prevent circularity (since we are estimating root zone storage capacity). In other words, there should be no water balance | 1G |

| | | Why the authors do selected monthly datasets and then performed downscaling with ERA-Interim? Why not using directly ERA-Interim data? Or other daily products (e.g., GLEAM for evaporation and TMPA for precipitation)? All these points should be clarified. | type of computation process involved in the determination of Sr. We used satellite-based evaporation products because they are the only options available that fulfill these criteria, i.e., reanalyses and land surface model evaporation contain soil depth information. Flux net data are too sparse for acquiring consistently good quality global coverage). The monthly satellite-based evaporation data used in the manuscript were those available at the time of this research. Conversely, precipitation data do not need to be satellite-based, but can also be ground-based. Inter-comparison of precipitation products show that both CRU and CHIRPS are good quality precipitation products. In particular, CHIRPS performance stands out in a comprehensive inter-comparison of 13 difference precipitation products in the Nile basin Hessels (2015)." | |
| | | Moreover, why the average of the three evaporation datasets should be "attractive"? Are the results changing by using only one of the datasets? What is the relative impact of the evaporation and precipitation datasets on the final results? | Added: "Nevertheless, data uncertainties still persist….[]… The use of three evaporation datasets decrease uncertainties related to individual evaporation products, because there is simply not one single preferred model. To compare the effect of different input data, we also present results of $S_R$ based on the separate evaporation and precipitation data in the Supplement." | 1G |
| | | Why ERA-Interim data are used for temporal downscaling? Apart that it is not mentioned how the temporal downscaling is carried out, currently daily evaporation and precipitation datasets are (freely) available (actually, several datasets). (1G)

Page 15, line 8 ff: please describe at least in some sentences the temporal downscaling of monthly evaporation and precipitation data.(2S) | Added "In the temporal downscaling, we first established the ratios between daily values to the mean monthly ERA-Interim, and second, used the relationship to estimate daily values from monthly $E_{SM}$ or $E_{CSM}$ values." | 1G, 2S |

| 3.2 Other data used in analyses | P16, L13: ERA-I evaporation is used as forcing of STEAM. What is the output of STEAM? Is it the actual evaporation? It should be clarified and clearly distinguished from potential evapotranspiration throughout the text. (1S)

page 16, line 14: "wind speed in two directions" really? If so, why do you use both directions? Or do you use the two measurements to calculate wind speed? (AK) | Modified to:
"Input ERA-I data to STEAM were at 3 h and 1.5 degree resolution and include: precipitation, snowfall, snowmelt, temperature at 2m height, dew point temperature at 2m height, wind speed vector fields (zonal and meridional components) at 10m height, incoming shortwave radiation, net long-wave radiation, and evaporation (only used to scale potential evaporation from daily to 3 h)." | 1S |
| | P16, L27-28: The methods used for downscaling/upscaling the different datasets should be described. | Modified to:
"Data with finer resolution than 0.5 degree have been upscaled to 0.5 degree by averaging (i.e., assuming that the value of a 0.5 degree grid cell correspond to the mean of the overlapping finer grid cell values). Data with coarser resolution than 0.5 degree were downscaled simply by transferring grid cell values (i.e., assuming that the finer 0.5 degree grid cell values correspond to those overlapped by the coarser degree grid cell values)." | 1S |
| | | The look-up table based $S_{R,steam}$ is derived by weighting rooting depth of a land cover type with the land cover type fractional area coverage in each grid cell. | N/A |
| | | | |
| **4. Results and discussion** | | | |
| **4.1 Root zone storage capacity estimates** | Page 40, Fig 3: For Africans desert region, the CRU-SM product has obviously values of 0, whereas the CHIRPS-CSM product is > 0, and if I see it correctly at Fig 3c, it has reasonable large values. Could you explain somewhere, where the difference comes from - is it due to precip product or due to additional evaporation product? | Added:
"The positive values of $S_{R,CHIRPS-CSM}$ in the Sahara desert are caused by overestimation of evaporation in the CMRSET evaporation product, (see also the Supplement)." | 2S |
| | P17, L12: "SR estimated are generally larger", larger than? Please clarify. | Revised to: "generally large" | 1S |

| | page 18, lines 12-28: In the discussion of the differences, it is also important to note that these datasets may use different climate data sets, particularly precipitation. Also, Kleidon (2004) calculated evaporation in a quite simple way, which also is likely to result in differences. What this means is that the differences may not simply reflect on different ways to infer rooting properties, but there is also a component related to the forcing datasets which is difficult to quantify. | Added:

"Nevertheless, different input data were also used in the different studies. Thus, it is difficult to attribute the variations in root zone storage capacity estimates to differences in methods contra differences in input data." | AK |
|---|---|---|---|
| **4.3 Implementation in a hydrological model** | P19, L22-P20, L8: Too many details are given here for the description of the differences of the simulated evaporation data. It is difficult to follow, please reduce the text focusing on the most relevant differences. | Modified:
"Figure 6 compares the STEAM-simulated evaporation when using, on the one hand, $S_{R,CRU-SM}$ and, on the other, the look-up table based $S_{R,STEAM}$. In general, $S_{R,CRU-SM}$ tend to lead to higher evaporation rates in the tropics and lower evaporation in the subtropics and temperate zone. In particular, the differences are pronounced during the warm and dry seasons. For example, the evaporation reductions with $S_{R,CRU-SM}$ is widespread in the Northern Hemisphere during its summer month July. During the dry seasons (e.g., January in the Sahel, July in Congo south of the Equator), the evaporation increase is the most significant. Moreover, the change in evaporation also depend on land cover type. In South America, evaporation increases in the seasonal tropical forests of the Amazon, whereas evaporation decreases in the savannas and shrublands in the south. These results suggest that $S_{R,CRU-SM}$ has the greatest potential to influence model simulations for the hot and dry seasons, in regions where the root zone storage varies strongly." | 1S |
| **4.4 The effect of different drought return periods** | | Updated return periods for land cover types after changes in results after changing to van Genuchten equation. | N/A |

| 4.5 Limitations | P22, L20: Recent studies have obtained huge differences between global scale precipitation datasets (e.g., Trenberth et al. (2014), Herold et al. (2016)). It seems not true that evaporation data (on a global scale) have larger spread than precipitation data. Please reformulate. | Modified:
"Finally, the quality of the estimated SR is dependent on the quality of the input evaporation and precipitation data. In this study, the choice of remotely sensed evaporation products influenced the resulting SR more than the choice of precipitation product, see the Supplement. In particular, the largest standard deviations in the ensemble evaporation products are located in central South America, the Sahel, India, and northern Australia (see Fig. 2e, 2f). To reduce uncertainty, the presented method is preferably applied using ensemble products based on reliable evaporation and precipitation datasets identified in comparison and evaluation studies (e.g., Bitew & Gebremichael, 2011; Herold, Alexander, Donat, Contractor, & Becker, 2015; Hessels, 2015; Hofste, 2014; Hu, Jia, & Menenti, 2015; Moazami, Golian, Kavianpour, & Hong, 2013; Trambauer et al., 2014; Trenberth et al., 2013; Yilmaz et al., 2014)" | 1S |
|---|---|---|---|
|  |  | Added:
"Finally, while the Sr estimates are model independent, the analyses of the best performing drought return periods of different land cover types will depend on the hydrological model used, given the large variations of evaporation estimates (and in particular transpiration/evaporation ratios) among land surface models (e.g., Wang and Dickinson 2012). Thus, although the contrasting return periods for woody land cover types and annual short vegetation types are supported by current knowledge about ecohydrological response to droughts, the calculated values are subject to assumptions. Uncertainties are probably largest for heterogeneous land cover types (such as savannahs) because they tend to be challenging to parameterise and simulate. Therefore, implementation of | N/A |

| | | Sr in other hydrological or land surface models would require model-specific analyses of optimal return periods." | |
|---|---|---|---|
| | | Added paragraph on the uncertainties relating to sun-sensor geometry in satellite-observed vegetation index. | N/A |
| | | | |
| **5. Summary and conclusion** | Page 23, line 2: after "from" can be misinterpreted and is not complete. I suggest to write: ": : :from remotely sensed evaporation, remotely sensed and station based precipitation and model based irrigation…" | Modified: "This study presents a method to estimate root zone storage capacity in principal from remotely sensed evaporation and observation-based precipitation data, by assuming that plants do not invest more in their roots than necessary to bridge a dry period." | 2S |
| | | Updated return periods for land cover types after changes in results after changing to van Genuchten equation. | N/A |
| | page 25, line 2: Note that for the effect of climate change, it also depends on the ability of vegetation to adapt to altered conditions. This aspect should be mentioned. | Added "…depending on the adaptability of vegetation to altered conditions." | AK |
| | | | |
| **References** | Page 28, line 12: belongs "Open Access" really to the journal title? | Corrected | 2S |
| | Page 20, line 14 f: Hard to judge that because de Boer-Euser is not available for the reviewers. Maybe you should write in a few sentences what is written there. | de Boer-Euser et al., (in review) is now published and the reference is updated accordingly. | 2S |
| | Page 31, line 3: Please check the citation. It is a master thesis, and I am not sure if there are so many co-authors. | Corrected | 2S |
| | Page 31, line 22: Check names | Corrected | 2S |
| | Page 32, line 4: does Jennings really have CMHCMH as initials? I tried to get access but that failed. Could you maybe update the resource? | Corrected | 2S |
| | Page 33, line 24: soil should be in lower case | Corrected | 2S |
| | Page 35, line 8: check initials from last co-author | Corrected | 2S |

| | | | |
|---|---|---|---|
| | page 7, line 10: I think more relevant is here a link to cost-benefit type of analysis rather than evolution. It may be appropriate to refer to the classic book edited by Givnish "On the economy of plant form and functioning" by Cambridge University Press. | Added GIvnish 1986 | AK |
| | | Added Döll 2003, Campos 2016, Hessels2015, Bitew2011, Herold2015, Moazami2013, Trenberth2013, vanGenuchten1980, Poulter2012,Hicke2007, Larson2001, Loehle1988, Wang2012 | N/A |
| | | Removed Hanasaki, Werth and Guntner. | |
| | | | |
| **Appendices** | P25, L13: It seems that "and SRCRU·SM" is missing here. | Added | 1S |
| | | | |
| **Figures** | Most of the figures are very tiny, and sometimes due to the choice of color very hard to read (e.g. Fig. A1). Please take care of figure size in the final production phase of the manuscript if it is accepted. | Changed color bars to discrete scale and enlarged Fig. 6. | 2G |
| | Tables/Figures: Please check captions for symbols. Captions should be selfdescribing. | Changes of captions in all tables and all figures. | 1S |
| | Page 13,line 3: I wonder how many of the 1.5 deg grid cells are available for each land cover class, if land cover needs to be at least 90% of a single land cover. Maybe I have misinterpreted the information, but is it correct that you used MODIS land cover with 0.05 resolution to assess land cover for the 1.5 cell? So, the 1.5 cell consists of 90 0.05 tiles, and at least 81 of them needs to be in one land cover class. The global pattern of land cover is very heterogeneous and think it is important for interpreting the results if you write (e.g. in a table) the | Added number of grid cells to Fig 9. | 2S |

| | | | |
|---|---|---|---|
| | number of grid cells per land cover class that went into that analysis.

Page 16, line 24: again, it would be nice to have an idea, how many grid cells are used per land cover class. | | |
| | Page 44, Fig 7: please write after aridity index that the calculation can be found in Sect B1. | Added :
"(defined in Appendix B1)" | 2s |
| | page 40, Figure 3: Start the caption more descriptive with something like "Estimates of root zone storage capacity of the …". | Added:

"Root zone storage capacity estimates of…" | AK |
| | page 40, Figure 3: You may also want to use the same color scale in panel (c) as in Fig. 4 to facilitate comparison? | Changed color scale to 550 in Fig 3 | AK |
| | page 43, Figure 6: I find the differences difficult to see. It may be easier to attribute the differences when you use only a few discrete color values with less than 8 shadings so that one more clearly associate the differences in a region with the values. (also applies to other plots) | Changed to discrete colorbar in Fig 2, 3, 4, 6, and A1. | AK |
| | page 38, Figure 1: This figure nicely illustrates the concept. I think it could be made even better if you show the integrated fluxes of Fin-Fout over time in a separate plot above the panel where you show the bins. | Figure changed. | AK |
| | | Updated evaporation figures 6, 7, 9 after changing to van Genuchten formulation. | N/A |
| | | | |
| **Supplementary Information** | Moreover, why the average of the three evaporation datasets should be "attractive"?
Are the results changing by using only one of the datasets? What is the relative impact of the evaporation and precipitation datasets on the final results? | New figures added to the Supplementary Information. | 1G |

| | | Comparison showing the differences between STEAM using the Matsumoto function and the van Genuchten function with root zone storage capacity. | N/A |
|---|---|---|---|
| | page 18, lines 4-11/Figure 4: the correspondence (or disagreements) between the data sets would be easier to see in a scatterplot, where the different data sets are compared at a grid-by-grid scale. How well they correspond is then reflected by the slope of the regression as well as the r2 value. It is probably not necessary to show all scatterplots (or add them as supplementary), but I think this type of analysis would really help to identify how well the different data sets compare to each other. | Added scatter plots and $R^2$ and RMSE comparisons. | AK |
| | | Correct variable name in Fig S4 | N/A |
| | | Re-plot S5, S6, for discrete colorbar. Table S1 for optimisation method. Updated Table S2 with new return periods. Table S3 for updated numbers. Update de Boer-Euser reference. | N/A |
| | | New merged Sr map and new STEAM run with SR,merged. | N/A |
| **No revisions made** | | | |
| **General** | "irrigation is captured in satellite-based evaporation data", I am not sure it is true. At least, not for all satellite-based datasets, please clarify. Moreover, at page 14 it reads that the evaporation originating from irrigation water simulated by LPJmL is considered. Why irrigation is already included in the satellite-based evaporation data? | No changes made since this is already explained in the manuscript. | 1G |
| | I suggest changing the first word of the title to Global-scale. That is more reflecting the issue, that the product is done for global land surface (but one can also live with just "global") | Kept "global" | 2S |
| | Page 8, line 10: change "global" by "global-scale" | Kept "global" | 2S |

| | | | |
|---|---|---|---|
| | Page 9, line 15 f: strange sentence. Why should one understand that irrigation is included in precipitation data? Or do you refer to satellite precipitation data? This sentence needs to be revised | Misunderstanding. | 2S |
| | Page 23, line 7: did you compare SR,chirps-CSM with E_SM or with E_CSM? Is that conclusion somewhere covered in the paper? | Already covered in the paper. | 2S |
| | page 14, line 14: "electro-magnetic spectrum". do you mean different wavelengths/bands? | "electro-magnetic spectrum" is correct. | AK |
| **Typesetting** | References, general: why are the page numbers written after the reference? Is this the new standard of HESS? | Will check this with the editorial staff. | 2S |

[revised manuscript text omitted]

$$f(\theta) = \frac{(\theta - \theta_{\mathrm{wp}})(\theta_{\mathrm{fc}} - \theta_{\mathrm{wp}} + c)}{(\theta_{\mathrm{fc}} - \theta_{\mathrm{wp}})(\theta - \theta_{\mathrm{wp}} + c)} \frac{\theta - \theta_{\mathrm{wp}}}{\theta_{\mathrm{fc}} - \theta_{\mathrm{wp}}}, \tag{6}$$

where $\theta$ is the actual volumetric soil moisture content (dimensionless), $\theta_{\mathrm{wp}}$ is the volumetric soil moisture content at wilting point, $\theta_{\mathrm{fc}}$ at field capacity. (This soil moisture stress  function departs from the original formulation in STEAM (Matsumoto et al., 2008; Wang-Erlandsson et al., 2014), which is described in Sect. 2 in the Supplementary Information.)

However, the root zone storage capacity $S_{\mathrm{R}}$ is simply location-bound (depending on climatic variables alone) and no longer considered a land cover and soil based parameter. Thus, to use $S_{\mathrm{R}}$ directly, we do not account for soil moisture below wilting point and assume $S_{\mathrm{R}} = h(\theta_{\mathrm{fc}} - \theta_{\mathrm{wp}})$, where $h$ is the rooting depth (m). The reformulated stress function of soil moisture becomes:

$$f(S) = \frac{S(S_{\mathrm{R}} + C)}{S_{\mathrm{R}}(S + C)} \frac{S}{S_{R}}, \tag{7}$$

where $S$ is the actual root zone storage (m). This reformulation is possible since the stress function retains its shape. Thus, $S_{\mathrm{R}}$ can in similar ways be implemented in other hydrological models.

To measure improvement, the root mean square error ($\varepsilon_{\mathrm{RMS}}$) for simulated evaporation is calculated using the original look-up table based root zone storage capacity $S_{\mathrm{R,STEAM}}$ and the newly derived root zone storage capacity  $S_{\mathrm{R,new}}$ (i.e., $S_{\mathrm{R,CRU-SM}}$ or $S_{\mathrm{R,CHIRPS-CSM}}$) respectively. The root mean square error improvement ($\varepsilon_{\mathrm{RMS,\,imp}}$) is positive if the $E$ simulated using $S_{\mathrm{R}}$ is closer to a benchmark evaporation data set than the $E$ simulated using $S_{\mathrm{R,STEAM}}$. The equation below shows the $\varepsilon_{\mathrm{RMS,imp}}$ of  $S_{\mathrm{R,new}}$:

$$\varepsilon_{\mathrm{RMS,\,imp}} = \varepsilon_{\mathrm{RMS}}(E_{S_{\mathrm{R,STEAM}}}, E_{\mathrm{benchmark}}) - \varepsilon_{\mathrm{RMS}}(E_{S_{\mathrm{R,new}}}, E_{\mathrm{benchmark}}).$$

The remote sensing based ensemble evaporation product $E_{SM}$

$$\varepsilon_{\mathrm{RMS, imp}} = \left[ \varepsilon_{\mathrm{RMS}}(E_{S_{\mathrm{R,STEAM}}}) - \varepsilon_{\mathrm{RMS}}(E_{\mathrm{SM}}) \right] -$$

$$\left[ \varepsilon_{\mathrm{RMS}}(E_{S_{\mathrm{R,CRU\text{-}SM}}}) - \varepsilon_{\mathrm{RMS}}(E_{\mathrm{SM}}) \right].$$

[revised manuscript text omitted]